

**Distinct diurnal variation of organic aerosol hygroscopicity and its relationship with**
**oxygenated organic aerosol**
**Ye Kuang[1,*,†],Yao He[2,†], Wanyun Xu[5], Yele Sun[2,*],Pusheng Zhao[6], Yafang Cheng[4], Gang Zhao[3],**
**Jiangchuan Tao[1], Nan Ma[1], Hang Su[4], Yanyan Zhang[1], Jiayin Sun[7],Peng Cheng[7], Wenda Yang[7],**
**Shaobin Zhang[1], Cheng Wu[7], Chunsheng Zhao[3]**
[1]{Institute for Environmental and Climate Research, Jinan University, Guangzhou 511443, China}
[2] {State Key Laboratory of Atmospheric Boundary Layer Physics and Atmospheric Chemistry,
Institute of Atmospheric Physics, Chinese Academy of Sciences, Beijing, China}
[3] {Department of Atmospheric and Oceanic Sciences, School of Physics, Peking University, Beijing,
China}
[4] {Max Planck Institute for Chemistry, Mainz 55128, Germany}
[5] {State Key Laboratory of Severe Weather & Key Laboratory for Atmospheric Chemistry, Institute
of Atmospheric Composition, Chinese Academy of Meteorological Sciences, Beijing, 100081, China}
[6] {Institute of Urban Meteorology, China Meteorological Administration, Beijing, 100089, China}
[7] {Institute of Mass Spectrometer and Atmospheric Environment, Jinan University, Guangzhou
510632, China}
† These authors contribute equally to this paper.
*Correspondence to: Ye Kuang (kuangye@jnu.edu.cn), Yele Sun (sunyele@mail.iap.ac.cn)



**Abstract**

The hygroscopicity of organic aerosols (OA) is important for investigation of its climatic and
environmental impacts. However, the hygroscopicity parameter $\kappa_{Org}$ remains poorly characterized,
especially in the relatively polluted environment on the North China Plain (NCP). Here we conducted
simultaneous wintertime measurements of bulk aerosol chemical compositions of $PM_{2.5}$ and $PM_1$ and
bulk aerosol hygroscopicity of $PM_{10}$ and $PM_1$ on the NCP using a capture vaporizer time-of-flight
aerosol chemical speciation monitor (ToF-ACSM) and a humidified nephelometer system which
measures aerosol light scattering enhancement factor $f(RH)$. A method for calculating $\kappa_{Org}$ based
on $f(RH)$ and bulk aerosol chemical composition measurements was developed. We found that $\kappa_{Org}$
varied in a wide range with significant diurnal variations. The derived $\kappa_{Org}$ ranged from almost 0.0
to 0.25 with an average (±1σ) of 0.08 (±0.06) for the entire study. The derived $\kappa_{Org}$ was highly
correlated with $f_{44}$ (fraction of $m/z$ 44 in OA), an indicator of oxidation degree of OA ($R$=0.79), and
the relationship can be parameterized as $\kappa_{Org} = 1.04 \times f_{44}$ - 0.02. On average, $\kappa_{Org}$ reached the
minimum (0.02) in the morning near 07:30 and then increased rapidly reaching the peak value of 0.16
near 14:30. The diurnal variations of $\kappa_{Org}$ were highly and positively correlated with those of mass
fractions of oxygenated OA ($R = 0.95$), indicating that photochemical processing played a dominant
role for the increase of $\kappa_{Org}$ in winter on NCP. Results in this study demonstrate the potential wide
applications of humidified nephelometer system together with aerosol composition measurements for
investigating the hygroscopicity of OA in various environments, and highlight that the
parameterization of $\kappa_{Org}$ as a function of OA aging processes needs to be considered in chemical
transport models for better evaluating the impacts of OA on cloud formation, atmospheric chemistry
and radiative forcing.


**1 Introduction**

Aerosol hygroscopic growth plays significant roles in different atmospheric processes including
atmospheric radiation transfer, cloud formation, visibility degradation, atmospheric multiphase
chemistry and even air pollution health effects, and therefore is crucial for studies on aerosol climatic
and environmental impacts. Organic materials in ambient aerosol particles, usually referred to as



organic aerosol (OA), contribute substantially to ambient aerosol mass and frequently contribute more
than half to submicron aerosol particles mass under dry state (Jimenez et al., 2009). The hygroscopicity
parameter κ (Petters and Kreidenweis, 2007) of organic aerosols ($\kappa_{Org}$) is a key parameter for
investigating the roles of organic aerosol in radiative forcing, cloud formation and atmospheric
chemistry. Liu and Wang (2010) demonstrated that ±50% changes in κ of secondary organic aerosol
(0.14±0.07) can lead to 40% changes in predicted cloud condensation nuclei (CCN) concentration.
Rastak et al. (2017) reported a global average difference in aerosol radiative forcing of -1 $W/m^2$
between $\kappa_{Org}$ of 0.05 and 0.15, which shares the same order with the overall climate forcing of
anthropogenic aerosol particles during the industrialization period. Li et al. (2019) reported that
organic aerosol liquid water contributed 18-32% to total particle liquid water content in Beijing.
Despite its importance, $\kappa_{Org}$ has not yet been well characterized due to the extremely complex
chemical compositions of organic aerosol. Therefore, it is important to conduct more researches on the
spatiotemporal variation and size dependence of $\kappa_{Org}$ and its relationship with aerosol chemical
compositions to reach a better characterization and come up with more appropriate parameterization
schemes in chemical, meteorological and climate models.

The large variety in OA chemical constituents makes it difficult to directly link $\kappa_{Org}$ to specific

organic aerosol compositions. The OA chemical composition is tightly connected to their volatile
organic precursors, which are also rich in variety and come from many natural and anthropogenic
sources. OA with different oxidation levels will also behave differently in respect to hygroscopic
growth. Thus, studies on $\kappa_{Org}$ at different locations and time periods have reported distinct
characteristics. Many studies have investigated the influence of OA oxidation level (represented by
O:C ratio or fraction of m/z 44 ACSM ion signal, $f_{44}$) on its hygroscopicity (Chang et al., 2010;Lambe
et al., 2011;Duplissy et al., 2011;Mei et al., 2013b;Wu et al., 2013;Hong et al., 2015;Chen et al.,
2017;Massoli et al., 2010) and have reached a conclusion that in average $\kappa_{Org}$ generally increases as
a function of organic aerosol oxidation level, however, the statistical empirical relationship between
$\kappa_{Org}$ and O:C ratio or $f_{44}$ differs much among different studies. Several studies have also analyzed
the diurnal variation characteristics of $\kappa_{Org}$ at different locations and periods (Cerully et al.,
2015;Bougiatioti et al., 2016;Deng et al., 2018;Deng et al., 2019;Thalman et al., 2017), with some
exhibiting distinct diurnal variations (Deng et al., 2018;Deng et al., 2019;Bougiatioti et al., 2016) and



others not so much (Cerully et al., 2015). Studies on $\kappa_{Org}$ has already been reported for several
locations around the world, however, only Wu et al. (2016) have reported the influences of OA
oxidation level on $\kappa_{Org}$ in the North China Plain (NCP) region, which is one of the most polluted
regions on earth. The diurnal characteristics of $\kappa_{Org}$ in the NCP have not been reported so far.
Therefore, more investigation into the diurnal variation of $\kappa_{Org}$ and its relationship to OA oxidation
level is required to better understand its characteristics in the NCP.

In addition, in previous studies on $\kappa_{Org}$, the Humidity Tandem Differential Mobility Analyzer

(HTDMA) or CCN counter were applied for aerosol hygroscopicity measurements. Both the HTMDA
and size-resolved CCN measurements can only be used to derive a $\kappa$ within a certain size range
(HDTMA: usually diameter below 300 nm, with a reported highest diameter of 360 nm (Deng et al.,
2019), CCN: with diameter up to ~200 nm (Zhang et al., 2014;Rose et al., 2010)). The aerosol particles
contributing most to aerosol optical properties (Bergin et al., 2001;Quinn et al., 2002;Cheng et al.,
2008;Ma et al., 2011;Kuang et al., 2018) and aerosol liquid water contents (Bian et al., 2014) in
continental regions are usually in the diameter range of 200 nm to 1μm, which the HTDMA and CCN
hygroscopicity measurements cannot represent. Results from several studies have reported that $\kappa_{Org}$
usually differentiates among particle size (Frosch et al., 2011;Kawana et al., 2016;Deng et al., 2019).
Especially, results from Deng et al. (2019) demonstrated that $\kappa_{Org}$ increase with the increase in
particle dry diameter. These results further demonstrate that studies about information of $\kappa_{Org}$ of
larger particles would be helpful for $\kappa_{Org}$ studies.

Other than HTDMA and CCN counter, the humidified nephelometer system which measures

aerosol light scattering enhancement factors is also widely used in aerosol hygroscopicity research
(Titos et al., 2016). The hygroscopicity parameter $\kappa$ retrieved from measured light scattering
enhancement factor is usually referred to as $\kappa_{f(\mathrm{RH})}$ (Chen et al., 2014;Kuang et al., 2017), which
represents the overall hygroscopicity of aerosol particles with their diameters ranging from 200 nm to
800 nm for continental aerosols (see discussions in Sect.3.3 for physical understanding of $\kappa_{f(\mathrm{RH})}$).
Using the retrieved $\kappa_{f(\mathrm{RH})}$ together with the according bulk aerosol chemical compositions
measurements of $PM_1$ (particulate matter with aerodynamic diameter less than 1 μm, corresponding
to mobility diameter of about 800 nm), $\kappa_{Org}$ can be derived, representing the hygroscopicity of
organic aerosol particles in the diameter range of 200 to 800 nm. In this study, both the light scattering
enhancement factor of $PM_{10}$ (particulate matter with aerodynamic diameter less than 10 μm) and $PM_1$



particles were measured. The aerosol chemical compositions were measured using an aerosol chemical
speciation monitor (ACSM). With these two aspects of aerosol measurements, $\kappa_{Org}$ is derived, and
the relationship between $\kappa_{Org}$ and the OA oxidation degree, as well as the diurnal variation of $\kappa_{Org}$
are investigated.

Site and instrument information are introduced in Sect.2. Method of deriving $\kappa_{Org}$ based on

retrieved $\kappa_{f(RH)}$ and bulk aerosol chemical compositions measurements are proposed and discussed
in Sect.3. Results and discussions are presented in Sect.4, followed by conclusions.
**2 Site and instruments**

From 11$^{th}$ November to 24$^{th}$ December 2018, continuous measurements of physical, optical and

chemical properties of ambient aerosol particles as well as meteorological parameters such as
temperature, wind speed and direction and relative humidity were made at the Gucheng site in
Dingxing county, Hebei province, China. This site is an Ecological and Agricultural Meteorology
Station (39°09'N, 115°44'E) of the Chinese Academy of Meteorological Sciences. The site locates
between Beijing (~ 100km) and Baoding (~40km), two large cities on the North China Plain, and is
surrounded by farmland and small residential towns.
**2.1 Inlet system and instruments**

During this field campaign, all instruments were housed in an air-conditioned container, with the

temperature held almost constant near 24 ℃. The schematic diagram of the inlet systems for the
aerosol sampling instruments is displayed in Fig.1. Three inlet impactors are used for aerosol sampling,
two PM$_{10}$ inlets and one PM$_1$ inlet, respectively sampling ambient aerosol particles with aerodynamic
diameter less than 10 μm and 1 μm. Nafion driers with lengths of 1.2 m were placed downstream of
each PM impactor inlet, which can drop the RH of sampled air below 15%, thus, sampled aerosol
particles can be treated as in dry state. Additionally, downstream every PM impactor inlet an MFC
(mass flow controller) and a pump was added for automatic flow compensation, to ensure that each
impactor reaches their required flow rate of 16.7 L/min and guaranteeing for the right cut diameters.



Aerosol sampling instruments can be categorized into four groups according to their inlet routes.
The first group (group1) downstream of the first PM10 inlet is comprised of only one instrument, the
Aerodynamic Particle Sizer (APS, TSI Inc., Model 3321), measuring the size distribution of ambient
aerosol particles with aerodynamic diameter ranging from 700 nm to 20 μm at a temporal resolution
of 20 seconds. The second group (group 2) includes a humidified nephelometer system (consisting of
two nephelometers and a humidifier) that measures aerosol optical properties (scattering and back

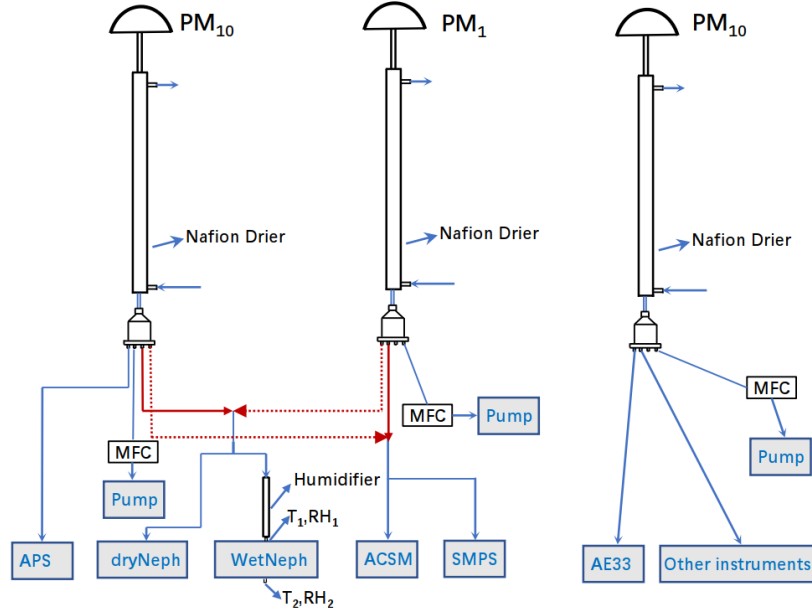

**Figure 1**. Schematic diagram of the inlet systems for aerosol sampling instruments

scattering coefficients at three wavelengths: 450 nm, 525 nm, 635 nm) of ambient aerosol particles in
dry state (DryNeph) and under 85% RH condition (WetNeph). The third group (group3) includes two
instruments, an ACSM and a scanning mobility particle sizer (SMPS; TSI model 3080). The ACSM
measures non-refractory particulate matter (NR-PM) species including organics, $SO_4^{2-}$, $NO_3^-$, $NH_4^+$,
and $Cl^-$ with an air flow of 0.1 L/min and a time resolution of 2 minutes. Since the ACSM` instrument
comes with an PM$_{2.5}$ impactor, chemical composition of PM$_{2.5}$ rather than of PM$_{10}$ were measured.
The SMPS measures particle mobility diameter size distributions with a diameter range of 12 nm
to 760 nm. The inlets of group2 and group3 switches every 15 minutes, as denoted by the dashed and
solid red lines in Fig.1, enabling the instruments of these two groups to alternately measure the
chemical and optical properties of PM$_{10}$ and PM$_1$. The fourth group (group4) includes an AE33


aethalometer (Drinovec et al., 2015) and other aerosol instruments. Due to technical issues with the
humidifier, the humified nephelometer system started to operate continuously since the 30[th] Nov .

In addition, the ambient meteorological parameters like RH, temperature, wind speed and

direction and air pressure were observed using an automatic weather station at a 1 minute time
resolution.
**2.2 The humidified nephelometer system**

The humidified nephelometer system we built was set up to measure dry state aerosol optical

properties at a fixed RH of 85%. The RH of the air sample is increased by a humidifier that consists
of two layers. The inner layer is a Gore-Tex tube layer passing through sampled air, while the outer
layer is a stainless-steel tube with circulating liquid water. The water vapor penetrates through the
Gore-Tex tube and humidifies the sample air, while liquid water is kept from the inner layer by the
Gore-Tex material. Upon the switch of inlets between group 2 and group 3, delays in valve switching
caused instantaneous low pressure in the sample air, which broke the humidifier with the Gore-Tex
tube after four days of continuous operation (3th, Dec) and flooded the WetNeph. The WetNeph was
fixed and recalibrated and a commercial Nafion drier (60 cm long, Perma Pure company) replaced the
Gore-Tex tube, which works the same way but is more resistant to low pressure. The temperature of
the circulating water layer is controlled by a water bath and specified by an algorithm that adjusts the
water temperature to maintain a relatively constant RH in the sensing volume of the WetNeph. To
monitor the RH in the sensing volume of the WetNeph, two temperature and RH sensors (Vaisala
HMP110, with accuracies of $\pm 0.2\,℃$ and $\pm 1.7\,\%$ for RH between 0 to 90%, respectively, and
$\pm 2.5\,\%$ for RH between 90 to 100%) were placed at the inlet and outlet of the WetNeph. Defining
measured RHs and temperatures at the inlet and outlet of the WetNeph as $RH_1/T_1$ and $RH_2/T_2$, the
according dew point temperatures $T_{d1}$ and $T_{d2}$ can be calculated and the average value $\overline{T_d}$ was treated
as the dew point of the sample air in the sensing volume of WetNeph. The sample RH is calculated
using $\overline{T_d}$ and the sample temperature measured by a sensor inside the sample cavity of the
nephelometer.


### 2.3 ACSM measurements and data analysis

The mass concentration and chemical composition of NR-PM species were measured with the Aerodyne ToF-ACSM which is equipped with a $PM_{2.5}$ aerodynamic lens (Williams et al., 2010) and a novel capture vaporizer (CV) (Xu et al., 2017;Hu et al., 2017) to extend the measured particle size to 2.5 μm. Detailed instrument descriptions were given in Fröhlich et al. (2013) and Xu et al. (2017). The ToF-ACSM data were analyzed with the standard data analysis software (Tofware v2.5.13, https://sites.google.com/site/ariacsm/) within Igor Pro (v6.37, WaveMetrics, Inc., Oregon, USA). A collection efficiency (CE) of 1 was used for all aerosol species, because the capture vaporizer has been proven to reach a unit CE for ambient aerosols (Hu et al., 2017;Hu et al., 2018b). Relative ionization efficiencies (RIE) of 3.06 and 1.09 were used for ammonium and sulfate quantification respectively, and the default values of 1.1 and 1.4 were used for nitrate and organic aerosol (OA) respectively. Compared with the AMS with standard vaporizer, the CV-ToF-ACSM reports higher fragments at small $m/z$'s due to additional thermal decomposition associated with increased residence time and hot surface collisions (Hu et al., 2018a). As a result, $f_{44}$ from CV-ToF-ACSM measurements is often much higher than those previously reported from AMS, yet they are well correlated (Hu et al., 2018a).

The organic mass spectra from $m/z$ 12 to 214 were analyzed by positive matrix factorization (PMF) (Paatero and Tapper, 1994) with an Igor Pro based PMF evaluation tool (v3.04) (Ulbrich et al., 2009). The $m/z$'s of 38, 49, 63 and 66 were removed from both $PM_1$ and $PM_{2.5}$ PMF inputs considering their small contributions to the total organic signal and their high noise. The PMF results were then evaluated following the procedures detailed in Zhang et al. (2011). After carefully evaluating the mass spectral profiles, diurnal patterns and temporal variation of the OA factors and comparing them with other collocated measurements, a five-factor solution was selected for both $PM_1$ and $PM_{2.5}$. The five factors include four primary factors, i.e., hydrocarbon-like OA (HOA), cooking OA (COA), biomass burning OA (BBOA), and coal combustion OA (CCOA), and a secondary factor, oxygenated OA (OOA). More detailed descriptions on the PMF results will be given in He et al. (in preparation).

### 2.4 Data reprocessing

The size distributions measured by APS were converted to mobility-equivalent size distributions using spherical shape assumptions and an effective particle density of 1.7 g/cm$^3$. Note that the





designations of PM$_{10}$ and PM$_1$ are in respect to aerosol aerodynamic diameters, while the
corresponding mobility-equivalent cut diameters of the two impactors are approximately 7669 nm and
767 nm, respectively. For simplicity and consistency, we will continue to refer to them as the PM$_{10}$ and
PM$_1$ based on their aerodynamic diameter. For the case of PM$_1$ measurements, the mobility-equivalent
cut diameter is quite near the upper range of the SMPS size range. Considering that the cut diameter
of the impactor corresponds to the diameter of aerosol particles in ambient state (aerosol hygroscopic
growth effect needs to be taken into account) and the SMPS measures the size distributions of aerosol
particles in dry state, the SMPS measurements should be able to cover the full size range of PM$_1$. When
the SMPS was sampling aerosol particles of PM$_{10}$, the size distributions measured by SMPS and APS
was merged together and truncated to an upper limit of 7669 nm to provide full range particle number
size distributions (PNSD). In addition, the AE33 measures aerosol absorption coefficient at several
wavelengths, the mass concentrations of black carbon (BC) were converted from measured aerosol
absorption coefficients at 880 nm with a mass absorption coefficient of 7.77 m$^2$/g (Drinovec et al.,

2015).

Since group 2 and 3 switched between PM$_1$ and PM$_{10}$ inlets every 15 minutes, all measurements
were averaged over each 15 minute observation episode, resulting in valid time resolutions of 15
minutes for APS and BC PM$_{10}$ measurements and 30 minutes for SMPS, ACSM and the humidified
nephelometer system PM$_1$ and PM$_{10}$ measurements, respectively. This resulted in a 15-minute time lag
between the averaged datasets of group 2 and group 3. To match the time of all the measurement data,
the measurements of SMPS, ACSM and the humidified nephelometer system were linearly
interpolated to the 15-minute time resolution of the APS data.
**3 Methodology**
**3.1 Calculations of hygroscopicity parameters $\kappa_{sca}$ and $\kappa$ from measurements of the**
**humidified nephelometer system**
The humidified nephelometer system measures aerosol light scattering coefficients and
backscattering coefficients at three wavelengths under dry state and 85% RH condition, providing
measurements of the light scattering enhancement factor $f(\text{RH}, \lambda)$, which is defined as





$f(\text{RH} = 85\%, \lambda) = \frac{\sigma_{sp}(RH, \ \lambda)}{\sigma_{sp}(dry \ \ \lambda)}$, with $\lambda$ being the light wavelength. In this study, we only calculate
$f(\text{RH}, 525\ nm)$ and refer to it hereinafter as $f(\text{RH})$ for simplicity. Brock et al. (2016) proposed a
single parameter formula to describe $f(\text{RH}, \lambda)$ as a function of RH. Kuang et al. (2017) further
developed this parameterization scheme to better describe measured $f(\text{RH})$ by including the
reference RH ($\text{RH}_0$) in the dry nephelometer as shown in Eq.1, using which the optical hygroscopicity
parameter $\kappa_{sca}$ can be derived from $f(\text{RH})_{measured}$.
$f(\text{RH})_{measured} = \left(1 + \kappa_{sca} \frac{RH}{100-RH}\right) / \left(1 + \kappa_{sca} \frac{RH_0}{100-RH_0}\right)$    (1)
An overall hygroscopicity parameter $\kappa$ referred to as $\kappa_{f(\text{RH})}$ can be retrieved from measured
$f(\text{RH})$ with the addition of simultaneously measured particle number size distribution (PNSD) and
BC mass concentration (Chen et al., 2014;Kuang et al., 2017). The idea is to conduct an iterative
calculation using the Mie theory and the κ-Köhler theory together to find a $\kappa_{f(\text{RH})}$ that closes the gap
between the simulated and the measured $f(\text{RH})$. Details on the calculations of $\kappa_{f(\text{RH})}$ can be found
in Kuang et al. (2017).
**3.2 Calculations of $\kappa_{chem}$ from aerosol chemical composition measurements**
For the calculation of aerosol hygroscopicity parameter $\kappa$ based on measured chemical
composition data ($\kappa_{chem}$), detailed information on the chemical species are needed. The ACSM can
only provide bulk mass concentrations of $SO_4^{2-}$, $NO_3^-$, $NH_4^+$, $Cl^-$ ions and organic components. For
the inorganic ions, a simplified ion pairing scheme (as listed in Tab.1) was used to convert ion mass
concentrations to mass concentrations of corresponding inorganic salts (Gysel et al., 2007;Wu et al.,

2016).

**Table 1**. Densities (ρ) and hygroscopicity parameters (κ) of inorganic salts used in this study

| Species | $NH_4NO_3$ | $NH_4HSO_4$ | $(NH_4)_2SO_4$ | $NH_4Cl$ |
|---|---|---|---|---|
| ρ (g $cm^{-3}$) | 1.72 | 1.78 | 1.769 | 1.527 |
| κ | 0.58 | 0.56 | 0.48 | 0.93 |

Mass concentrations of $SO_4^{2-}$, $NO_3^-$, $NH_4^+$, $Cl^-$ are thus specified into ammonium sulfate (AS),
ammonium nitrate (AN) ammonium chloride (AC) and ammonium bisulfate (ABS), with the $\kappa$ values
of these salts specified according to (Wu et al., 2016) and Liu et al. (2014) (Tab.1). For a given internal





mixture of different aerosol chemical species, a simple mixing rule called Zdanovskii–Stokes–
Robinson (ZSR) can be used for predicting the overall $\boldsymbol{\kappa_{chem}}$ on the basis of volume fractions of
different chemical species ($\varepsilon_i$) (Petters and Kreidenweis, 2007):
$\kappa_{chem} = \sum_i \kappa_i \cdot \varepsilon_i$        (2)
where $\kappa_i$ and $\varepsilon_i$ represent the hygroscopicity parameter κ and volume fraction of chemical
component $i$ in the mixture. Based on Eq.2, $\kappa_{chem}$ can be calculated as follows:
$\kappa_{chem} = \kappa_{AS}\varepsilon_{AS} + \kappa_{AN}\varepsilon_{AN} + \kappa_{ABS}\varepsilon_{ABS} + \kappa_{AC}\varepsilon_{AC} + \kappa_{BC}\varepsilon_{BC} + \kappa_{Org}\varepsilon_{Org}$    (3)
where $\kappa_{Org}$ and $\varepsilon_{Org}$ represent κ and volume fraction of total organics. Since black carbon is
hydrophilic, $\kappa_{BC}$ is assumed to be zero. With known $\boldsymbol{\kappa_{chem}}$, $\kappa_{Org}$ can be calculated using the
following formula:
$\kappa_{Org} = \dfrac{\kappa_{chem} - (\kappa_{AS}\varepsilon_{AS} + \kappa_{AN}\varepsilon_{AN} + \kappa_{ABS}\varepsilon_{ABS} + \kappa_{AC}\varepsilon_{AC})}{\varepsilon_{Org}}$        (4)
To calculate volume fractions of individual species, their volume concentrations and the total volume
concentration of aerosol particles ($V_{tot}$) are required. The volume concentration of salts can be
calculated from the additive ion mass concentrations divided by their densities listed in Tab.1. The
volume concentration of organics was calculated by assuming density of POA as 1 g/cm$^3$ and density
of OOA as 1.4 g/cm$^3$ (Wu et al., 2016). For the calculation of $V_{tot}$, we have three choices. The first
choice is to sum up the volume concentrations of all chemical species (AS, AN, ABS, AC, BC and
organics), where the volume concentration of BC was calculated by assuming a density of 1.7 g/cm$^3$.
We refer the calculated total volume concentration of aerosol particles to as $V_{tot,Chem}$. The second
choice is to   integrate $V_{tot}$ from measured PNSD, using the equation $V_{tot,PNSD} = \int \frac{4}{3}\pi r^3 n(r) dr$,
where r is the particle radius and n(r) is the measured PNSD. The third choice is to use the trained
machine learning estimator to estimate the $V_{tot}$ based on measurements of the dry nephelometer
($V_{tot,Neph}$) as was introduced in Kuang et al. (2018). $V_{tot}$ of PM$_1$ calculated using these three methods
were compared to each other and shown in Fig.S2. $V_{tot,Chem}$ correlates well with $V_{tot,PNSD}$, but in is
on average 30% lower than that of $V_{tot,PNSD}$. Chemical components within aerosol particles such as
dust, sea salt as well as metal ions could not be identified by ACSM. Since the Gucheng site is far from
the ocean, sea salt should have negligible impacts on the total mass of PM$_1$. However, mineral dust

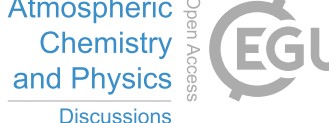

can extend into the submicron range (Shao et al., 2007), which might be the cause for the underestimate
in total mass concentration calculated using ACSM and BC data. $V_{tot,Neph}$ also correlates well with
$V_{tot,PNSD}$, but is on average 16% lower than that of $V_{tot,PNSD}$. Closure studies between modelled and
measured $\sigma_{sp}$ and $\sigma_{bsp}$ at 525 nm for PM$_1$ and PM$_{10}$ aerosol particles all showed good agreement
between theoretical modelling results and measurements (Fig.S1), with most points falling between
the 20% relative deviation lines. However, modelled $\sigma_{sp}$ for both PM$_1$ and PM$_{10}$ were obviously
higher than measured $\sigma_{sp}$, with an average relative difference of 22% and 13% between them for
PM10 and PM1, respectively. The result for PM$_1$ explains why $V_{tot,Neph}$ was lower than $V_{tot,PNSD}$.
Two reasons might have contributed to this discrepancy: (1) both PNSD and aerosol optical property
measurements carry non-negligible uncertainties, with the SMPS bearing measurement uncertainty of
30% for particles larger than 200 nm, which contribute most to $V_{tot}$ (Wiedensohler et al., 2012), and
the nephelometer measured $\sigma_{sp}$ having an uncertainty of 9% (Sherman et al., 2015;Titos et al., 2016);
(2) The sampling tube length, valves, tube angles and flow rates are different for the dry nephelometer
and SMPS (e.g. the tube length is much shorter and flow rate smaller for SMPS than those for the dry
nephelometer), leading to different wall loss and loss in semi-volatile of aerosol components. ACSM
and the dry nephelometer had similar tube length and nephelometer measurements bears less
uncertainty than SMPS. Thus, $V_{tot,Neph}$ was chosen as $V_{tot}$ in the calculations of Eq.4. Based on the
calculated $V_{tot}$, the material unidentified by ACSM accounts for 19% of $V_{tot}$ on average, could not
be neglected in the $\kappa_{Org}$ calculation. Thus, Eq.4 was modified as follows:
$\kappa_{Org} = \dfrac{\kappa_{chem} - (\kappa_{AS} \cdot \varepsilon_{AS} + \kappa_{AN} \cdot \varepsilon_{AN} + \kappa_{ABS} \cdot \varepsilon_{ABS} + \kappa_X \cdot \varepsilon_X)}{\varepsilon_{Org}}$     (5)
where $\kappa_X$ and $\varepsilon_X$ are hygroscopicity parameter $\kappa$ and volume fractions of the unidentified material.
Previous studies using $V_{tot,Chem}$ as the total volume concentration of aerosol particles have avoided
the discussion about influences of unidentified material by the ACSM or other aerosol mass
spectrometer instruments. The hygroscopicity of these unidentified materials, which might be dust or
other components in continental regions, were not discussed before. Dust is nearly hydrophilic, with
mineral dust showing $\kappa$ in range of 0.01 to 0.08 (Koehler et al., 2009). In this paper, we arbitrarily
specified $\kappa_X$ to be 0.05.



### 3.3 Can $\kappa_{f(\text{RH})}$ represent $\kappa_{chem}$?

According to Eq.5, the measured bulk $\kappa_{chem}$ values are needed to derive $\kappa_{Org}$. Bulk aerosol chemical compositions and aerosol hygroscopicity $\kappa_{f(\text{RH})}$ measurements are available, one would naturally jump to the conclusion of treating $\kappa_{f(\text{RH})}$ as $\kappa_{chem}$ to derive $\kappa_{Org}$ (both are from bulk aerosol measurements). However, the relationship between $\kappa_{chem}$, $\kappa_{f(\text{RH})}$ and the size-resolved $\kappa$ distribution needs to be clarified in order to answer the question whether $\kappa_{f(\text{RH})}$ can accurately represent $\kappa_{chem}$.

Using $V_i$ to represent volume concentrations of chemical species $i$ and $V_i(D_p)$ to represent volume concentrations of species $i$ with particle diameter of $D_p$, $\kappa_{chem}$ can be derived as follows based on Eq.2,:

$$\kappa_{chem} = \sum_i \kappa_i \cdot \varepsilon_i = \sum_i \frac{V_i}{V_{tot}} \cdot \kappa_i = \sum_i \frac{1}{V_{tot}} \cdot \int \frac{d\,V_i\,(D_p)}{dlogD_p} \cdot dlogD_p \cdot \kappa_i. \tag{6}$$

By swapping the order of summation and integration, Eq.6 can be written as:

$$\kappa_{chem} = \int \frac{1}{V_{tot}} \cdot \sum_i \frac{d\,V_i\,(D_p)}{dlogD_p} \cdot dlogD_p \cdot \kappa_i. \tag{7}$$

Considering that $\kappa_{D_p} = \sum_i \frac{dV_i\,(D_p)}{dV(D_p)} \cdot \kappa_i$, Eq.7 can be rewritten as:

$$\kappa_{chem} = \frac{1}{V_{tot}} \int \kappa_{D_p} \cdot dV(D_p) \tag{8}$$

Result of Eq.8 indicates that $\kappa_{chem}$ calculated using Eq.3 represents the overall hygroscopicity of aerosol particles with volume contribution as the weighting function of $\kappa_{D_p}$.

As for $\kappa_{f(\text{RH})}$, a detailed analysis is performed here to facilitate its physical understanding. The differential form of $\sigma_{sp}$ of aerosol particles in dry state can be expressed as follows:

$$\sigma_{sp} = \int \frac{d\sigma_{sp}}{dlogD_p} dlogD_p \tag{9}$$

Based on the definition of $f(\text{RH})$, $\sigma_{sp}$ of aerosol particles under different RH conditions can be written as:

$$\sigma_{sp}(RH) = \int \frac{d\sigma_{sp}}{dlogD_p} \cdot f_{D_p}(RH) \cdot dlogD_p \tag{10}$$

Therefore, the differential form of observed overall $f(\text{RH})$ can be formulated as:





$f(\text{RH}) = \int \frac{1}{\sigma_{sp}} \cdot \frac{d\sigma_{sp}}{dlogD_p} \cdot f_{D_p}(\text{RH}) \cdot dlogD_p$   (11)
Based on this formula, the sensitivity of $f(\text{RH})$ on the hygroscopicity of aerosol particles with
diameter $D_p$ $(\kappa_{D_p})$ can be derived as:
$\frac{1}{dlogD_p} \cdot \frac{\partial f(\text{RH})}{\partial \kappa_{D_p}} = \frac{1}{\sigma_{sp}} \cdot \frac{d\sigma_{sp}}{dlogD_p} \cdot \frac{\partial f_{D_p}(\text{RH})}{\partial \kappa_{D_p}}$   .   (12)

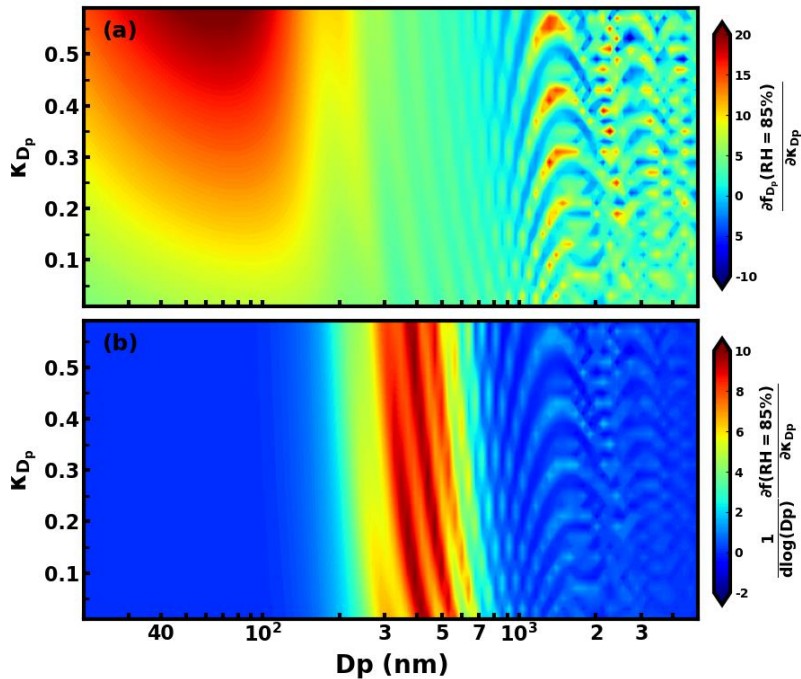

**Figure 2**. (a) simulated $\frac{\partial f_{D_p}(\text{RH})}{\partial \kappa_{D_p}}$; (b) simulated $\frac{1}{dlogD_p} \cdot \frac{\partial f(\text{RH})}{\partial \kappa_{D_p}}$

The sensitivity of $f(\text{RH})$ to $\kappa_{D_p}$ are determined together by the two terms in Eq. 12: (1) $\frac{1}{\sigma_{sp}} \cdot \frac{d\sigma_{sp}}{dlogD_p}$,
which represents the contribution of $\sigma_{sp}$ of aerosol particles in dry state with diameter $D_p$ to total
$\sigma_{sp}$, and (2) $\frac{\partial f_{D_p}(\text{RH})}{\partial \kappa_{D_p}}$, which represents the sensitivity of $f_{D_p}(\text{RH})$ to $\kappa_{D_p}$. Based on the Mie theory
and the κ-Köhler theory, we simulated the second term under 85% RH condition for varying $D_p$ and
$\kappa_{D_p}$ values (Fig.2a). In the diameter range below 200 nm, $\frac{\partial f_{D_p}(\text{RH})}{\partial \kappa_{D_p}}$ is very high, displaying a
maximum near 60 nm. In this diameter range, larger $\kappa_{D_p}$ generally corresponded to higher $\frac{\partial f_{D_p}(\text{RH})}{\partial \kappa_{D_p}}$.


For $200\,\text{nm} < D_p < 800\,\text{nm}$, higher and lower $\frac{\partial f_{D_p}(RH)}{\partial \kappa_{D_p}}$ appear alternatively, with all values
staying positive. For $D_p > 800\,nm$, maxima and minima regions appear alternatively, and $f_{D_p}(RH)$
might decrease with increasing $\kappa_{D_p}$. This is because, at this diameter range, the aerosol scattering
efficiency has a non-monotonic response to the particle diameter increase (see Fig.2a of (Kuang et al.,

2018)).

The first term of Eq.9, representing size-resolved $\sigma_{sp}$ contributions of particles with diameter in

dry state, mainly depends on the PNSD. The average PNSD of $PM_{10}$ was applied in the simulation the
first term using Mie theory (Fig.S3). Combining results of the first term and second term, the sensitivity
of $f(RH)$ to $\kappa_{D_p}$ was obtained and depicted in Fig.2b. Results reveal that $f(RH)$ is quite sensitive
to the $\kappa_{D_p}$ of particles within 200 to 800 nm diameter range, but almost insensitive to $\kappa_{D_p}$ of particles
with diameters below 200 nm and above 800 nm (corresponding aerodynamic diameter of about 1
$\mu$m). For particles smaller than 200 nm, the first term was quite small especially for particles smaller
than 100 nm (Fig.S3), while for particles larger than 800 nm, in addition to a small first term, the
second term fluctuated between negative and positive values, which is why $f(RH)$ was not sensitive
to the overall hygroscopicity of these larger aerosol particles. These results suggest that although
$\kappa_{f(RH)}$ was derived from $f(RH)$ measurements of $PM_{10}$, it mainly represents the overall
hygroscopicity of aerosol particles with dry diameters between 200 and 800 nm for continental
aerosols. This result indicates that $\kappa_{f(RH)}$ derived from $f(RH)$ measurements of $PM_{10}$ and $PM_1$
should differ little from each other for measurements conducted in continental regions.

However, the quantitative relationship between $\kappa_{f(RH)}$ and size-resolved $\kappa_{D_p}$ is still not clear.

Based on Eq.11, $f_{D_p}(RH)$ can be expressed as:
$f_{D_p}(RH) = \frac{d\sigma_{sp}(RH)}{d\sigma_{sp}} = \frac{\frac{1}{4}\pi \cdot (D_p \cdot g)^2 \cdot Q_{sca}(D_p, g) \cdot dN}{d\sigma_{sp}},\,(13)$
where $g$ is the growth factor of aerosol particles which is a function of $\kappa_{D_p}$ and RH (Brock et al.,
2016), i.e. $g = (1 + \kappa_{D_p} \cdot \frac{RH}{100-RH})^{1/3}$, $dN$ is differential form of aerosol number concentration, and
$Q_{sca}$ is the scattering efficiency as a function of $D_p$ and $g$. The results of Kuang et al. (2018) indicate
that, under dry state, $Q_{sca}$ can expressed as $Q_{sca} = k \cdot D_p$ with $k$ varying as a function of $D_p$. Here,



we follow this idea and express the $Q_{sca}$ under humidified condition as $Q_{sca}(D_p, g) = C \cdot D_p \cdot g$,
where C is a function of $D_p$, $\kappa_{D_p}$ and RH. Replacing $g$ and $Q_{sca}$ in Eq.13, we yield:
$f_{D_p}(RH) = \dfrac{\frac{1}{4}\pi \cdot D_p{}^3 \cdot C(\text{Dp},\kappa_{D_p},\text{RH}) \cdot (1 + \kappa_{D_p} \cdot \frac{RH}{100 - RH}) \cdot dN}{d\sigma_{sp}}$,   (14)
which we can substitute into Eq.8, to obtain a new expression for $f(RH)$:
$f(RH) = \int \dfrac{\frac{1}{4}\pi \cdot D_p{}^3 \cdot C(\text{Dp},\kappa_{D_p},\text{RH}) \cdot (1 + \kappa_{D_p} \cdot \frac{RH}{100 - RH}) \cdot dN}{\sigma_{sp}}$  (15)
If we define $X_c(\text{Dp}, \kappa_{D_p}, \text{RH}) = C(\text{Dp}, \kappa_{D_p}, \text{RH})/k$, and considering that $d\sigma_{sp} = \frac{1}{4} \cdot \pi \cdot D_p{}^2 \cdot Q_{sca} \cdot$
$dN = \frac{1}{4} \cdot \pi \cdot D_p{}^3 \cdot k \cdot dN$, Eq.14 can be written as:
$f(RH) = \int \dfrac{X_c(\text{Dp},\kappa_{D_p},\text{RH}) \cdot (1 + \kappa_{D_p} \cdot \frac{RH}{100 - RH}) \cdot d\sigma_{sp}}{\sigma_{sp}}$  (16)
The $\kappa_{f(RH)}$ is a uniform $\kappa$ for aerosol particle sizes that can yield simulated $f(RH)$ equal to the
measured one. Thus, $f(RH)$ can also be expressed as:
$f(RH) = \int \dfrac{X_c(\text{Dp},\kappa_{f(RH)},\text{RH}) \cdot (1 + \kappa_{f(RH)} \cdot \frac{RH}{100 - RH}) \cdot d\sigma_{sp}}{\sigma_{sp}}$   (17)
Combining Eq.16 and Eq.17, the relationship between $\kappa_{f(RH)}$ and size-resolved $\kappa_{D_p}$ can be derived
as:
$\kappa_{f(RH)} = \dfrac{\int X_c(\text{Dp},\kappa_{D_p},\text{RH}) \cdot \kappa_{D_p} \cdot d\sigma_{sp}}{\int X_c(\text{Dp},\kappa_{f(RH)},\text{RH}) \cdot d\sigma_{sp}} + \dfrac{\int (X_c(\text{Dp},\kappa_{D_p},\text{RH}) - X_c(\text{Dp},\kappa_{f(RH)},\text{RH})) \cdot d\sigma_{sp}}{\int X_c(\text{Dp},\kappa_{f(RH)},\text{RH}) \cdot d\sigma_{sp}} \cdot \dfrac{100 - RH}{RH}$.   (18)
$X_c$ values under 85% RH for different $D_p$ and $\kappa_{D_p}$ values are simulated and shown in Fig.3, based





on which the second term of Eq.18 (which depends on the PNSD and size-resolved $\kappa_{D_p}$) could be

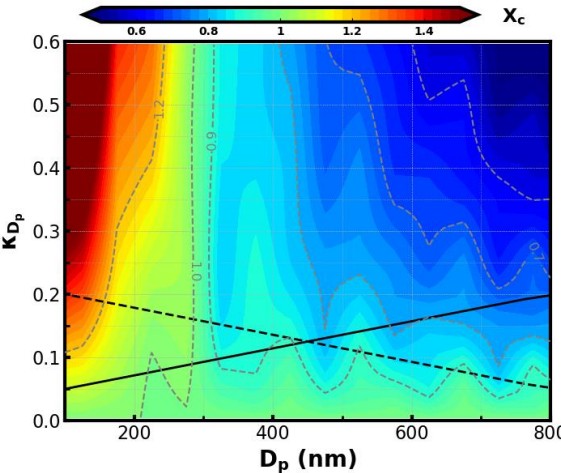

**Figure 3**. Simulated values of $X_c$ under 85% RH for different $D_p$ and $\kappa_{D_p}$ values. Black solid and dashed lines are two assumed size-resolved $\kappa_{D_p}$ distributions.

calculated using the average PNSD during this field campaign and two assumed extreme cases of size-
resolved $\kappa_{D_p}$ (solid and dashed black lines in Fig.3). For PM$_1$, the second term corresponding to the
two size-resolved $\kappa_{D_p}$ cases were -0.007 and 0.008, respectively. Corresponding values simulated for
PM$_{10}$ were -0.005 and 0.004, respectively. To further investigate the possible contribution range of the
second term to $\kappa_{f(\mathrm{RH})}$, size-resolved $\kappa_{D_p}$ derived by Liu et al. (2014) based on size resolved chemical
composition measurements in ambient atmosphere on the NCP region (Fig.S4) were used with the
average PNSD during this campaign to calculate values of the second term. Calculated values of
second term ranged from -0.005 to 0.009, with its contribution to $\kappa_{f(\mathrm{RH})}$ ranging from -1.5% to 2%


(0.3% on average). These results indicate that the second term was negligible in most cases, and Eq.18
could be approximated as:
$\kappa_{f(\text{RH})} \approx \dfrac{\int X_c(\text{Dp},\kappa_{D_p},\text{RH}) \cdot \kappa_{D_p} \cdot d\sigma_{sp}}{\int X_c(\text{Dp},\kappa_{f(\text{RH})},\text{RH}) \cdot d\sigma_{sp}}$    (19)
$X_c$ values shown in Fig.3 indicate that for aerosol particles in the diameter range of 200 to 800 nm
(which contribute most to $\sigma_{sp}$ and is the part of the aerosol population $\kappa_{f(\text{RH})}$ is most sensitive to)
and for the observed $\kappa_{D_p}$ range of continental aerosols ($\kappa_{D_p}$ usually less than 0.5), $X_c$ mainly ranged
from 0.7 to 1. Considering this, we might approximately assume $X_c$ in Eq.18 as a constant value.
Then, Eq.19 can be further simplified to:
$\kappa_{f(\text{RH})} \approx \dfrac{1}{\sigma_{sp}} \int \kappa_{D_p} \cdot d\sigma_{sp}$    (20)
This result suggests that $\kappa_{f(\text{RH})}$ can be approximately understood as the overall hygroscopicity of
aerosol particles with the $\sigma_{sp}$ contribution as the weighting function of $\kappa_{D_p}$.
Based on results of Eq.8 and 20, both $\kappa_{f(\text{RH})}$ and $\kappa_{chem}$ represent the overall hygroscopicity of
bulk aerosol particles, however, their weighting functions of $\kappa_{D_p}$ are different. Within a certain $D_p$

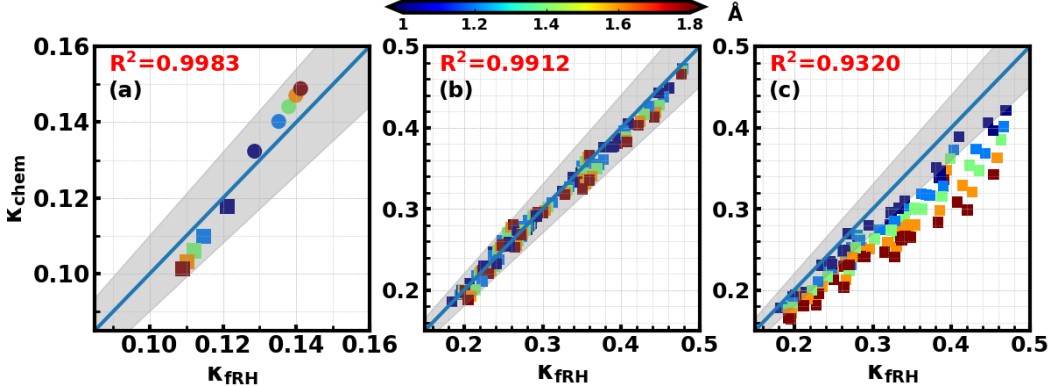

**Figure 4.** $\kappa_{chem}$ versus $\kappa_{f(\text{RH})}$, colors represent average Ångström exponent (Å) values of corresponding PNSD (a)corresponding to size-resolved $\kappa_{D_p}$ distributions shown in Fig.4; **(b)** and **(c)** corresponding to size-resolved $\kappa_{D_p}$ distributions shown in Fig.S4 for PM1 and PM10, respectively. Gray areas represent the absolute relative differences between $\kappa_{chem}$ and $\kappa_{f(\text{RH})}$ are less than 10%.

range, aerosol $\sigma_{sp}$ is approximately proportional to aerosol volume (Kuang et al., 2018), resulting in
little difference between $\kappa_{f(\text{RH})}$ and $\kappa_{chem}$. In this study, bulk $\kappa_{f(\text{RH})}$ was measured for both PM$_1$
and PM$_{10}$. How much does $\kappa_{chem}$ differ from $\kappa_{f(\text{RH})}$ for PM$_1$ and PM$_{10}$ samples? Both PNSD and
size-resolved $\kappa_{D_p}$ distributions contribute to the difference between $\kappa_{chem}$ and $\kappa_{f(\text{RH})}$. To study





their influences in a simple and apparent way, $\kappa_{chem}$ and $\kappa_{f(\mathrm{RH})}$ were simulated based on the two
extreme cases of size-resolved $\kappa_{D_p}$ distributions in Fig. 3 and five average PNSDs corresponding to
five ranges of aerosol Ångström exponent (0.9-1.1,1.1-1.3,1.3-1.5,1.5-1.7,1.7-1.9) during this field
campaign. In the instance of PM$_1$, as can be seen in Fig.4a, assuming a $\kappa_{D_p}$ increasing as a function
of $D_p$ resulted in $\kappa_{chem}$<$\kappa_{fRH}$ (square points in Fig.4a), especially for PNSDs corresponding to
larger Ångström exponents. This is because the volume contributions of small particles (e.g. particles
with $D_p$ between 100 to 300 nm) to $V_{tot}$ are larger than their light scattering coefficient contributions
to $\sigma_{sp}$ (as shown in Fig.S6), thus the hygroscopicity of small particles had larger impacts on $\kappa_{chem}$
than $\kappa_{fRH}$. Higher Ångström exponents generally correspond to shift in PNSD towards smaller $D_p$,
which exacerbates the contribution of small particles, further increasing the difference between $\kappa_{chem}$
and $\kappa_{fRH}$. For the case with $\kappa_{D_p}$ decreasing as a function of $D_p$ (circle markers in Fig.4a) it is vice
versa, resulting in $\kappa_{chem}$>$\kappa_{fRH}$. In general, for these two extreme cases of size-resolved $\kappa_{D_p}$
distributions, the absolute value of the relative difference between $\kappa_{chem}$ and $\kappa_{f(\mathrm{RH})}$ ranged from
2.8% to 7.5% with an average of 4.8%. This result indicates that for PM$_1$, $\kappa_{chem}$ might differ little
from $\kappa_{f(\mathrm{RH})}$ since $\kappa_{D_p}$ usually varies less with $D_p$ in ambient atmosphere than in the two assumed
cases (Liu et al., 2014). The average size-resolved $\kappa_{D_p}$ distribution from Haze in China campaign
(Liu et al., 2014) indicate that $\kappa_{D_p}$ varies significantly for $D_p$<250 nm, while it varies less within the
diameter range of 250 nm to 1 μm. To further study the variation range of the relative difference
between $\kappa_{chem}$ and $\kappa_{f(\mathrm{RH})}$ under ambient conditions, the size-resolved $\kappa_{D_p}$ distributions derived
from measured size-resolved chemical compositions in the NCP region (Liu et al., 2014) (shown in
Fig.S5) were used in simulations and results are shown in Fig.4b. The absolute value of the relative
difference between $\kappa_{chem}$ and $\kappa_{f(\mathrm{RH})}$ ranged from 0.04% to 8% with an average and standard
deviation of 2.8±2%, which further confirms that for PM$_1$ $\kappa_{f(\mathrm{RH})}$ can accurately represent $\kappa_{chem}$ in
most cases.

For PM$_{10}$, values of $\kappa_{chem}$ and $\kappa_{f(\mathrm{RH})}$ using $\kappa_{D_p}$ size distributions derived from ambient

measurements were simulated and displayed in Fig.4c. The simulated absolute values of the relative





difference between $\kappa_{chem}$ and $\kappa_{f(RH)}$ ranged from 0.2% to 41% with an average and standard
deviation of 16±8 %, with all $\kappa_{chem}$ lower than $\kappa_{f(RH)}$. This is because, for PM10, super-micron
particles typically with low hygroscopicity (Fig.S4) contribute much more to $V_{tot}$ than to $\sigma_{sp}$ (as
shown in Fig.S7). These results indicate that, for PM10, $\kappa_{f(RH)}$ cannot accurately represent $\kappa_{chem}$.

Above analysis results indicate that $\kappa_{f(RH)}$ retrieved from light scattering measurements of PM$_1$

represent accurately the $\kappa_{chem}$ of PM$_1$ and can be used in Eq.5 as measured $\kappa_{chem}$ for deriving $\kappa_{Org}$.
**4 Results and discussions**
**4.1 Overview of the campaign data**

The timeseries of ambient RH, chemical compositions of PM$_{2.5}$ and PM$_1$, $\sigma_{sp}$ at 525 nm of PM$_{10}$

and PM$_1$ in dry state, calculated $\kappa_{sca}$ and $\kappa_{f(RH)}$ values of PM$_{10}$ and PM$_1$ are shown in Fig.5. Overall,
the mass concentrations of NR-PM$_{2.5}$ and NR-PM$_1$ ranged from 1 to 221 $\mu g/cm^3$ and from 1.8 to
326 $\mu g/cm^3$, with average concentrations of 63 and 93 $\mu g/cm^3$, respectively. Measured $\sigma_{sp}$ at
525 nm of PM$_{10}$ and PM$_1$ ranged from 11 to 1875 $Mm^{-1}$ and from 18 to 2732 $Mm^{-1}$, with average
values of 550 and 814 $Mm^{-1}$, respectively. These results demonstrate that this campaign was carried
out at a site that is overall very polluted, quite clean conditions as well as extremely polluted conditions
were experienced during the measurement period. The mass contributions of ammonium, nitrate,
sulfate and organics to NR-PM$_{2.5}$ and NR-PM$_1$ are listed in Table 2, which show that on average
organics contributed most to the mass concentration of NR-PM$_1$ and NR-PM$_{2.5}$. During the first period,



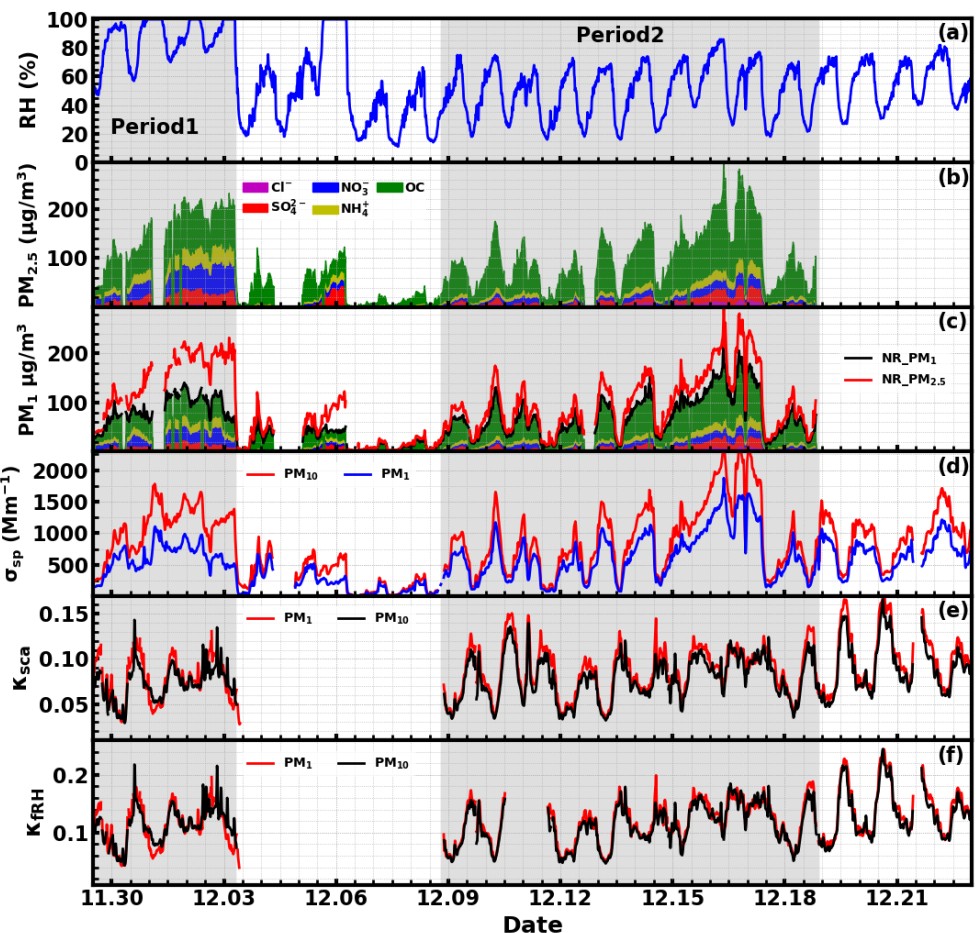

Figure 5. timeseries of ambient RH (a), chemical compositions of PM$_{2.5}$ (b) and PM$_1$ (c), $\sigma_{sp}$ at 525 nm of PM$_{10}$ and PM$_1$ (d), calculated $\kappa_{sca}$ (e) and $\kappa_{f(RH)}$ (f) values of PM$_{10}$ and PM$_1$.

nitrate contributed most to inorganics, while inorganics contribute most to mass concentrations of NR-
PM$_{2.5}$ and NR-PM$_1$. During the second period, the ambient RH is relatively lower than that of the first
period, ranging from 16% to 86% with an average of 49%. During this period, organics contributed
most to mass concentrations of NR-PM$_{2.5}$ and NR-PM$_1$, with the NR mass concentrations of PM2.5
and $\sigma_{sp}$ at 525 nm of PM$_{10}$ being only 33% and 40% higher than those of PM$_1$.
The time series of calculated $\kappa_{sca}$ and $\kappa_{f(RH)}$ are shown in Fig.5e-f. $\kappa_{sca}$ of PM$_1$ and PM$_{10}$
ranged in between 0.01 to 0.2 and 0.02 to 0.17, with corresponding averages of 0.09 and 0.08. From
near 12:00 on the 10$^{th}$ Dec to about 12:00 on the 11$^{th}$ Dec, the $\kappa_{f(RH)}$ was not available due to the
absence of PNSD measurements during that period. $\kappa_{f(RH)}$ of PM$_1$ and PM$_{10}$ respectively ranged



from 0.02 to 0.27 and from 0.03 to 0.26, with corresponding averages of 0.12 and 0.12. These results
indicate that the hygroscopicity during this campaign was generally low, which could be associated
with the high mass contributions of organics. The range as well as the average level of $\kappa_{f(RH)}$ is quite
consistent with the results obtained from another field campaign conducted at the same site in winter
2016, suggesting low aerosol hygroscopicity conditions in winter to be prevalent at this site.
Additionally, it can be noted that except for fog events, $\kappa_{sca}$ and $\kappa_{f(RH)}$ values of $PM_1$ are generally
higher than those of $PM_{10}$, however, with relative small differences (10% and 3.5% for $\kappa_{sca}$ and
$\kappa_{f(RH)}$, respectively). Although particles with diameters above 800 nm impact almost negligibly on
retrieved $\kappa_{f(RH)}$ (refer to discussions in Sect3.3), it can still cause a small difference between $\kappa_{f(RH)}$
of PM10 and PM1. Results of previous studies indicate that the overall hygroscopicity of aerosol
particles larger than 800 nm are usually low and are typically lower than the overall hygroscopicity of
accumulation mode particles (Liu et al., 2014), which may explain why $\kappa_{f(RH)}$values of $PM_1$ are
generally higher than those of $PM_{10}$ during non-fog periods.
**Table 2**. Average (range) mass contribution of ammonium, nitrate, sulfate and organics to NR-PM2.5 and NR-
PM1 during different periods.

| Species | Ammonium | | nitrate | | sulfate | | Organics | |
|---|---|---|---|---|---|---|---|---|
| | $PM_1$ | $PM_{2.5}$ | $PM_1$ | $PM_{2.5}$ | $PM_1$ | $PM_{2.5}$ | $PM_1$ | $PM_{2.5}$ |
| **Entire** | 12% | 12% | 13% | 14% | 10% | 11% | 59% | 59% |
| **period** | 0.2-24% | 0.1-24% | 2-31% | 1-32% | 0.3-49% | 0.2-50%% | 12-99% | 4-91% |
| **Period 1** | 15% | 16% | 22% | 24% | 13% | 14% | 47% | 42% |
| **Fog** | 10-17% | 12-18% | 11-28% | 16-30% | 9-15% | 12-16% | 30-65% | 37-55% |
| **Period 1** | 17% | 16% | 23% | 23% | 12% | 12% | 43% | 44% |
| **non-fog** | 10-22% | 7-21% | 6-31% | 5-32% | 8-23% | 7-17% | 32-75% | 31-69% |
| **Period 2** | 12% | 10% | 11% | 10% | 8% | 7% | 64% | 67% |
| | 0.2-20% | 0.1-19% | 5-30% | 4-29% | 0.3-16% | 0.2-16% | 40-82% | 40-85% |

During fog periods, a large part of submicron particles in dry state will activate into fog droplets,
which are super micron particles in ambient state (see PNSD example in Fig.S4a), exerting substantial
impacts on $f(RH)$ measurements of $PM_{10}$ which are not detectable in the $PM_1$ measurements. Since
for a certain particle diameter and fog supersaturation, particles with higher hygroscopicity are more



readily activated, the observed PM$_{10}$ $\kappa_{f(RH)}$ increased during fog events and often exceeded those of
PM$_1$ in contrast to non-fog periods (Fig.5f).

**4.2 $\kappa_{Org}$ derivations and its relationship with organic aerosol oxidation state**

The discussion results in Sect.3.3 demonstrate that $\kappa_{f(RH)}$ of PM$_1$ accurately represents $\kappa_{chem}$

in most cases, thus a closure study between calculated $\kappa_{chem}$ of PM$_1$ based on measured chemical
compositions and measured $\kappa_{chem}$ (represented by PM$_1$ $\kappa_{f(RH)}$) can be conducted using Eq.3 if $\kappa_{Org}$
were a known parameter. A $\kappa_{Org}$ of 0.06 was used in this closure test, which was the calculated by
Wu et al. (2016) based on aerosol chemical composition and aerosol hygroscopicity measurements.
The comparison between measured and calculated $\kappa_{chem}$ as shown in Fig.6a has not achieved very

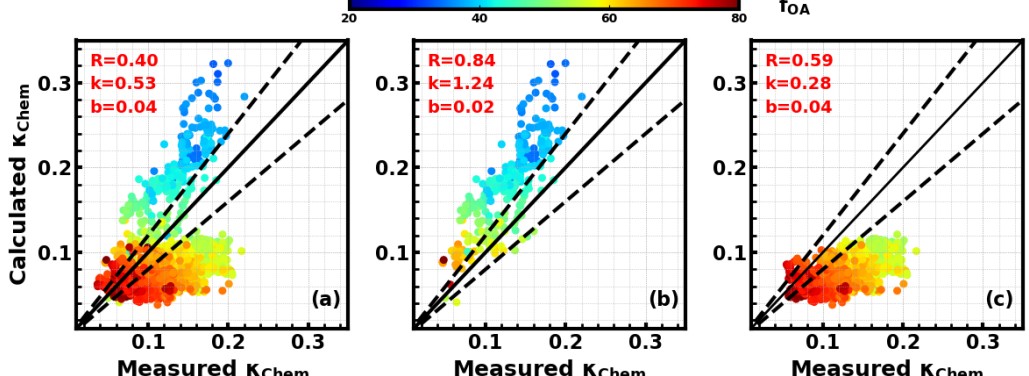

**Figure 6**. Comparison between measured and calculated $\kappa_{chem}$ by assuming a $\kappa_{Org}$ of 0.06. **(a)** The whole period; **(b)** Only Period 1; **(c)** Only Period 2. Colors represents the mass fractions of organic aerosol in NR-PM1 (f$_{OA}$), and the color bar is shown on the top.

good agreement. On one hand, calculated $\kappa_{chem}$ overestimated the measured one when mass fractions
of organic aerosol (f$_{OA}$) was lower than 45%, while on the other hand    calculated $\kappa_{chem}$
underestimated the measured one in most cases when f$_{OA}$ was higher than 45%. As introduced in
Sect.4.1, these two situations roughly correspond to situations of Period 1 and 2, respectively.
Separating the data points shown in Fig.6a into Periods 1 (Fig.7b) and 2 (Fig.7c), it can be seen that
all low f$_{OA}$ data points are found in Period 1, with most of the data points showing f$_{OA}$ less than 50%.
Although the calculated $\kappa_{chem}$ during this period was on average 25% higher than the measured
$\kappa_{chem}$, they were highly correlated (R=0.84). A similar case was also found in Wu et al. (2013), and
they conclude that the loss of ammonium nitrate (semi volatile particles) in the HTDMA might be the



reason. The relationship between nitrate concentration and the difference between calculated and
measured $\kappa_{chem}$ was investigated, which confirmed that the discrepancy was highly correlated to
mass fractions of nitrate in NR-PM1(Fig.S7), suggesting that the overestimation of calculated $\kappa_{chem}$
might be associated with the volatile loss of ammonium nitrate. Since the tube length (from the splitter
to inlet of instrument) of wet nephelometer was about 1 m longer than that of the ACSM, there probably
was more loss in ammonium nitrate in the wet nephelometer.

During Period 2, the average mass fraction of nitrate was low (11%), which is why the loss of

ammonium nitrate had little influence on $\kappa_{chem}$ estimations (Fig.S7). However, during Period 2,
when organic aerosol was the dominating, the calculated $\kappa_{chem}$ underestimated measured $\kappa_{chem}$ in
most cases (Fig.6c). Previous studies have shown larger $\kappa_{Org}$ for OA with higher oxidation level
(Chang et al., 2010;Duplissy et al., 2011;Wu et al., 2013), which might have contributed to the
underestimation in $\kappa_{chem}$. This gave us the hint that Period 2 might provide us with a good opportunity
to study $\kappa_{Org}$. Following the method in Sect. 3.2, $\kappa_{Org}$ was derived using Eq.5, resulting in a $\kappa_{Org}$

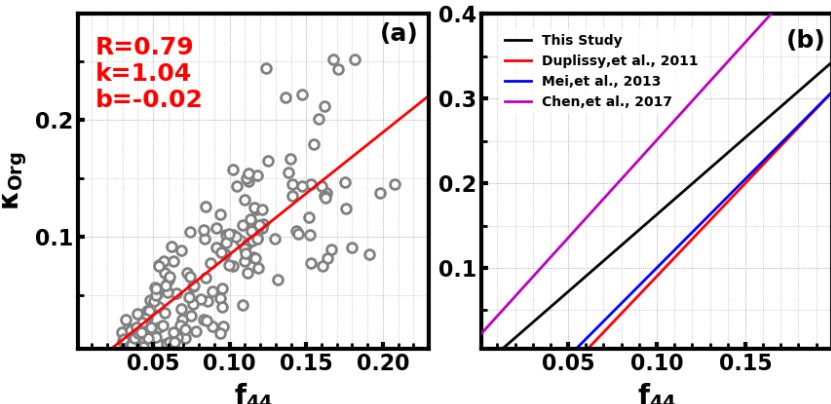

Figure 7. **(a)** the relationship between derived $\kappa_{Org}$ and $f_{44}$; **(b)** Comparison with earlier studies.

ranging from 0.0 to 0.25, with an average of 0.08±0.06. This indicates that using a constant $\kappa_{Org}$
value in the calculation of $\kappa_{chem}$ would result in large bias. To further investigate the impact of OA
oxidation level on $\kappa_{Org}$, we compared the derived $\kappa_{Org}$ against $f_{44}$, which is often used to represent
the oxidation level of OA. Results show a clear positive correlation (R=0.79) and a statistical
relationship of $\kappa_{Org} = 1.04 \cdot f_{44}$ - 0.02 (Fig.7a), indicating that the degree of oxidation level is a
crucial parameter determining the OA hygroscopicity. The derived empirical relationship between



522 $\kappa_{Org}$ and $f_{44}$ was compared to results in earlier studies (Fig.7b). As mentioned in Sect.2.3, $f_{44}$ from

523 CV-ToF-ACSM measurements is much higher than those previously reported from AMS, but they are

524 well correlated and the ratio between $f_{44}$ of CV-ToF-ACSM and previous AMS instruments for

525 ambient aerosol ranges from 1.5 to 2 with an average of 1.75. Therefore, to be consistent with the $f_{44}$

526 in previous studies, the empirical relationship in Fig.7b is changed to $\kappa_{Org} = 1.79 \cdot f_{44}$ - 0.03. The

527 $\kappa_{Org}$ values are lower than that from the scheme of Chen et al. (2017), but higher than those in

528 Duplissy et al. (2011) and Mei et al. (2013a). In general, results of all published studies demonstrate

529 that hygroscopicity of organic aerosol generally increase as the oxidation level of organic aerosol

530 increases, however, the empirical mathematical relationship differs much among different studies

531 (Hong et al., 2018). These results highlight that more studies are required to study the influence of OA

532 oxidation level on $\kappa_{Org}$ to approach a more universal parameterization scheme that can be used in

533 chemical, meteorological and climate models.

534 **4.3 Distinct diurnal variations of $\kappa_{Org}$ and its relationship with OOA**

535 The discussions in Sect. 4.2 already proved that $\kappa_{Org}$ was highly variable, which is why we need

536 to know its variational characteristics and influencing factors. The time series of derived $\kappa_{Org}$ is

537 depicted in Fig.8a, showing large $\kappa_{Org}$ fluctuations within a day. The average diurnal profile of $\kappa_{Org}$

538 (Fig.8b) displays a distinct diurnal variation, with $\kappa_{Org}$ reaching its minimum (0.02) in the morning

539 (near 07:30 LT) and increasing quickly to a maximum (0.19) near 14:30 LT. During daytime, the water

540 uptake abilities of organic aerosol particles changed from near hydrophobic to moderately hygroscopic

541 within 7 hours. Previous results from observations in Japan also revealed significant $\kappa_{Org}$ diurnal

542 variations, however, with daily minima in the afternoon hours due to the increase of less oxygenated

543 OA mass fractions (Deng et al. (2018) and Deng et al. (2019)). Such large variability and significant

544 diurnal variations of $\kappa_{Org}$ were observed for the first time on the NCP.



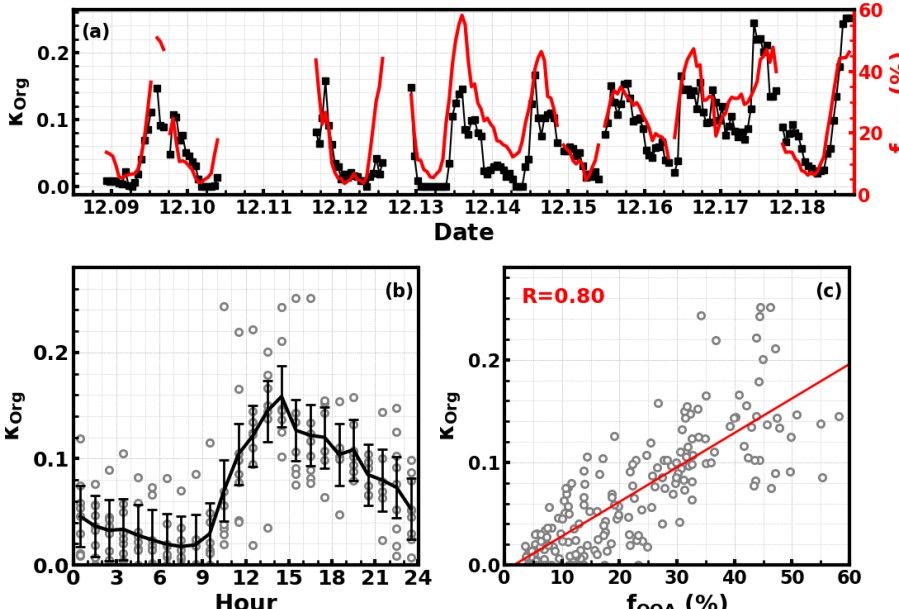

Figure 8. (a) Time series of derived $\kappa_{Org}$ and OOA mass fraction in NR-PM1 ($f_{OOA}$) in the right y-axis; (b) Average diurnal profile of $\kappa_{Org}$; (c) Scatter points of $\kappa_{Org}$ versus $f_{OOA}$ (%), and red line is the fitting line with linear regression.

Results introduced in Sect.4.2 demonstrated that $\kappa_{Org}$ was highly correlated to the OA oxidation
level, which suggests that $\kappa_{Org}$ might be associated with the oxygenated part of organic aerosol. In
this study, the mass concentrations of OOA were derived using PMF analysis, and its mass fraction in
the total organic aerosol mass ($f_{OOA}$) was calculated (Fig.8a). $f_{OOA}$ displays diurnal variations similar
to $\kappa_{Org}$ and the statistical relationship between $\kappa_{Org}$ and $f_{OOA}$ (Fig.8c) shows a strong correlation
(R=0.8), which both hint that OOA might be a determining factor for $\kappa_{Org}$.
The correlation coefficient between the average diurnal profiles of $\kappa_{Org}$ and $f_{OOA}$ was 0.95,
which suggests that the variations in $f_{OOA}$ was driving the significant diurnal variations of $\kappa_{Org}$. The
average diurnal variations of mass concentrations of identified OOA, HOA, COA, CCOA, BBOA, and
their mass fractions in total organic mass are shown in Fig.9a and Fig.9b, respectively. The mass
concentrations of HOA, CCOA and BBOA decreased sharply from the morning time to about 15:00 LT
due to boundary layer evolution processes. The mass concentrations of COA increased a little in the
morning and then decreased quickly after 09:30 LT. This transitory increase of COA in the morning
might be associated with the cooking for breakfast. However, the OOA mass increased sharply from





about 07:30 to 10:30 LT even under the quick boundary layer development during this time range,
remaining almost constant thereafter. The rapid decrease of primary organic aerosol components and
rise in OOA concentration resulted in dramatic increase of $f_{OOA}$ from 9:00 to 13:30 LT in the

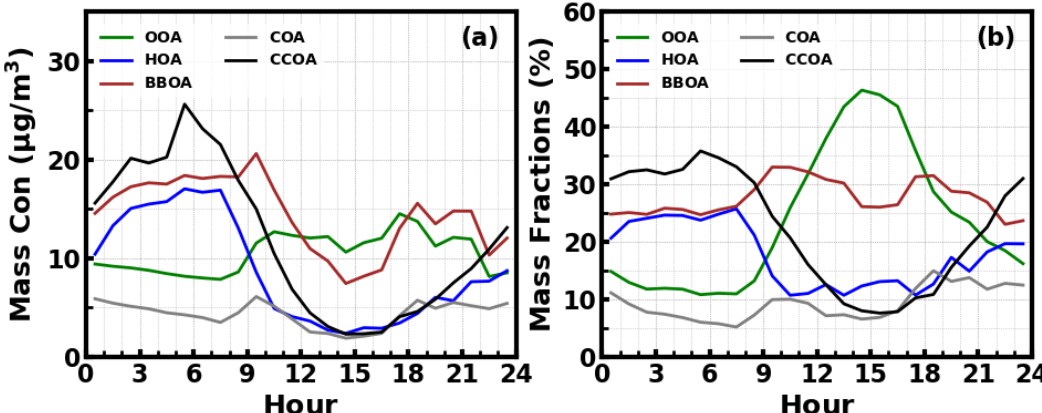

Figure 9. (a) Average diurnal profiles of mass concentrations of OOA, HOA, COA, CCOA, BBOA; (b) Average diurnal variations of mass fractions of OOA, HOA, COA, CCOA, BBOA.

afternoon, this period also corresponds to the fastest increase period of $\kappa_{Org}$. After 14:30 LT, the OOA
mass concentration varied little, however, mass concentrations of primary organic aerosol components
increased quickly, resulting in the decrease of $f_{OOA}$ and $\kappa_{Org}$.
**5 Conclusions**

A field campaign was conducted in winter 2018 on the North China Plain, using a humidified

nephelometer system and a ACSM to measure the bulk aerosol hygroscopicity of $PM_{10}$ and $PM_1$ and
bulk aerosol chemical compositions of $PM_{2.5}$ and $PM_1$.

During this field campaign, the air was highly polluted with high aerosol loadings. Measured $\sigma_{sp}$

at 525 nm of $PM_{10}$ and $PM_1$ in dry state ranged from 11 to 1875 $Mm^{-1}$ and from 18 to 2732 $Mm^{-1}$
with average values of 550 and 814 $Mm^{-1}$, respectively. Retrieved $\kappa_{f(RH)}$ of $PM_{10}$ and $PM_1$ ranged
between 0.02 to 0.27 and 0.03 to 0.26, with averages of 0.12 and 0.12, respectively. The $\kappa_{f(RH)}$
(derived from aerosol light scattering enhancement $f(RH)$) difference between $PM_{10}$ and $PM_1$ was
found to be relatively small (3.5% in average), which was consistent with the physical and
mathematical interpretation of $\kappa_{f(RH)}$.

A method of calculating $\kappa_{Org}$ (organic aerosol hygroscopicity) base on $f(RH)$ and bulk aerosol





chemical composition measurements is proposed. The key part of this method is that the size cut of
bulk aerosol chemical composition measurements should be $PM_1$ no matter the bulk $\kappa_{f(\mathrm{RH})}$ is
retrieved from light scattering enhancement measurements of $PM_1$ or $PM_{10}$. The derived $\kappa_{Org}$ ranged
from 0.0 to 0.25 with an average of 0.08, which highlights that $\kappa_{Org}$ displayed a large variability on
the NCP and that large uncertainties would rise if a constant $\kappa_{Org}$ were used to estimate the climatic
and environmental effects of organic aerosols. The variation of $\kappa_{Org}$ was significantly positively
correlated to the oxidation degree of organic aerosols. In addition, a distinct diurnal variation of $\kappa_{Org}$
was found, with a minimum in the morning (0.02) and maximum in the afternoon(0.16), indicating
that the organic aerosol changed from near hydrophobic to near moderately hygroscopic during
daytime within only 7 hours, which was observed for the first time in the NCP region. The distinct
diurnal variations of $\kappa_{Org}$ were associated with the significant diurnal variations of mass fractions of
oxygenated organic aerosol in total organic aerosol mass. The rapid formation of oxygenated organic
aerosol together with the dilution of primary organic aerosol during the development of the boundary
layer resulted in the quick increase of mass fractions of oxygenated organic aerosol and $\kappa_{Org}$.

The large variability and distinct diurnal variations in $\kappa_{Org}$ found in this study reveal the urgent

need for more studies on the spatial and temporal variations of $\kappa_{Org}$ in the NCP region to better
characterize $\kappa_{Org}$. The significant influences of organic aerosol aging processes on organic aerosol
hygroscopicity should be considered in studying roles of organic aerosol in cloud formation,
atmospheric radiative effects and atmospheric chemistry.

**Data availability**. The data used in this study are available from the corresponding author upon request
(kuangye@jnu.edu.cn) and (sunyele@mail.iap.ac.cn).

**Competing interests**. The authors declare that they have no conflict of interest.

**Author Contributions**. YK conceived and organized this paper. YC, HS, NM, YK and JT planned
this campaign. YK, YS and NM designed the experiments. YK and YH conducted the ACSM and
aerosol light scattering enhancement factor measurements. YZ and SZ conducted the particle number
size distribution measurements. JS and WY conducted the black carbon measurements. YH performed
the ACSM PMF analysis. WX, YH, YS, CZ, PZ and YC helped the data analysis, and WX helped



much in the language editing. YK, YH and YS prepared the manuscript with contributions from all co-
authors.

**Acknowledgments**
This work is supported by National Key Research and Development Program of China (Grant
2017YFC0210104),National Natural Science Foundation of China (91644218), the National research
program for key issues in air pollution control (DQGG0103) and the Guangdong Innovative and
Entrepreneurial Research Team Program (Research team on atmospheric environmental roles and
effects of carbonaceous species: 2016ZT06N263). We also thanks scientists and technicians from Max
Planck Institute for Chemistry, Mainz for supporting this field campaign.

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
