# Peer review of "Distinct diurnal variation of organic aerosol hygroscopicity and its relationship with"

_Atmospheric Chemistry and Physics, 2019_

## Referee Comment (RC1) · Anonymous Referee #1 · 13 Nov 2019

**General comments:**

This manuscript has introduced the simultaneous $f$(RH) and chemical measurements with a humidified nephelometer system and CV-ToF-ACSM for both $PM_1$ and $PM_{10}$ conducted in the wintertime of 2018 in the north China plain. The bulk hygroscopicity parameter, $\kappa$, results were calculated from both the light scattering growth measurements and chemical compositions. The two types of bulk $\kappa$ values were compared and discussed in detail, with a good comparison achieved for $PM_1$ measurements especially for polluted continental aerosol particles. Further, the authors innovatively proposed a new algorithm of deriving $\kappa_{org}$ from the $f$(RH) and chemical data, favoring the direct measure of the water uptake by organic fractions with commercially available instruments and conventional data set. A pronounced diurnal pattern for $\kappa_{org}$ was identified and presented for the first time in the northern China. The variation of $\kappa_{org}$ was closely related to both $f_{44}$ and $f_{OOA}$, signifying the importance of atmospheric oxidation processes on the water uptake by organic species. The findings reported in this study could serve as a good reference for modelling investigations on the climatic effects particularly driven by atmospheric organic aerosol particles in polluted continental regions.

The techniques used in this study are valid, and the reported hygroscopicity data are comparable to those in previous studies. In general, the quality of this manuscript is good yet could be improved, providing that some of the introduction and discussion contents (see the specific comments attached) were organized and delivered in a more logical/concise way. Also, some ambiguous expressions can be avoided, and a thorough grammar check is highly recommended. I would like to suggest its final publication upon a minor revision, with the comments specified as below.

**Specific comments:**

1. Page 3, line 58: *"... lead to **40% changes** in predicted cloud condensation nuclei (CCN) concentration."* The "*40% changes*" here is confusing, as which is difficult to tell from the sentence whether "*a 40% increase or decrease in the $N_{CCN}$*" was resulted from the *"50% increase or decrease in $\kappa_{org}$"*. The similar problem exists in the following sentence, which didn't state clearly the corresponding relationship between the average difference in aerosol radiative forcing and change in $\kappa_{org}$., e.g., which scenario ($\kappa_{org}$ =0.05 or $\kappa_{org}$ =0.15) corresponds to a higher radiative forcing? A straightforward delivery way is necessary to avoid ambiguity.

2. Page 3, line 65: Unlike the variation of $\kappa_{org}$ itself, I'm afraid I didn't find any connection between the **importance of size-dependent $\kappa_{org}$** and the above-mentioned content. Some details and corresponding references are needed to support the importance of the size influence on $\kappa_{org}$ and related climatic effects. I would recommend the authors to reorganize the context of size influence on $\kappa_{org}$, which can be combined with the information provided in the third paragraph (i.e., contents related to the HTDMA and CCN measurements).

3. Page 4, line 109: How is the '*mobility diameter of 800 nm*' obtained? Related information and references are preferred for the conversion here.

4. Page 6, line 151: Based on the introduction of each instrument, only the humidified nephelometer can measure both $PM_{10}$ and $PM_1$. How can chemical compositions of $PM_{10}$ be measured with ACSM, which is designed with a $PM_{2.5}$ aerodynamic lens/impactor as mentioned in Line 148? Similarly, how can SMPS measure the size distribution of $PM_{10}$?

5. Page 8, line 206: Why is the density for size conversion regarded as 1.7 g/cm$^3$, the same as that of black carbon used in the calculation of $\kappa_{org}$? According to the data reported, the organic fraction is always the predominant contributor to the particle mass of $PM_{2.5}$. This might suggest a smaller density for the ambient particles. Then how to evaluate the uncertainty in the related calculations?

6. Page 10, line 243: For the "*iterative calculation using the Mie theory*", how are the chemical composition and corresponding mixing state of particles considered in the $\kappa_{f(RH)}$ calculation? This would affect the closure/comparison between derived $\kappa_{f(RH)}$ and $\kappa_{chem}$, thus the interpretation of representativeness of $\kappa_{f(RH)}$.

7. Page 11, line 266: I suppose that you were assuming black carbon as hydrophobic, rather than hydrophilic, and $\kappa_{BC}$ is approximately taken as 0. Supporting references would be needed for this point and also for the density assumption of BC in Line 275. A similar typo was found for the description of 'Dust' in Page 12, line 309, which would be hydrophobic instead of hydrophilic.

8. Page 12, line 307: What does the 'by' mean: "… *influences of unidentified material by the ACSM …*"? Are you suggesting 'not detected by' ACSM?

9. Page 13, line 313: "*Bulk aerosol chemical compositions and aerosol hygroscopicity $\kappa_{f(RH)}$ measurements are available, one would naturally jump to the conclusion of treating $\kappa_{f(RH)}$ as $\kappa_{chem}$ to derive $\kappa_{Org}$ (both are from bulk aerosol measurements).*"

   A connection like a conjunction is needed for the whole sentence. "Both" here sounds ambiguous, although I would assume them to be $\kappa_{f(RH)}$ *and* $\kappa_{chem}$. It's better to specific which two hygroscopicity parameters you were referring to.

10. Page 16, Line 380: Why is the $\kappa_{f(RH)}$ uniform for all the particle sizes? If yes, does it mean that $\kappa_{Dp} = \kappa_{f(RH)}$, while $\kappa_{Dp}$ itself is size dependent?

11. Page 17, Line 387: What does the 'which' mean?

    Line 388: Is there any consideration of choosing "two extreme cases of size-resolved $\kappa_{Dp}$"?

12. Page 18, Figure 4(c): In comparison of the PM$_1$ scenario displayed in Fig.4b, $\kappa_{f(RH)}$ is generally higher than $\kappa_{chem}$, and larger discrepancies exist for the PM$_{10}$ case. Can you provide some hints for these results?

    Line 411: "*How much does $\kappa_{chem}$ differ from $\kappa_{f(RH)}$ for PM$_1$ and PM$_{10}$ samples?*" In my understanding, the $\kappa_{chem}$ of PM$_{10}$ samples is calculated from the corresponding chemical compositions that are actually measured for PM$_{2.5}$ instead of PM$_{10}$ (due to the configuration of ACSM with a PM$_{2.5}$ impactor), when sampling with a PM$_{10}$ inlet. In this sense, the two hygroscopicity parameters for PM$_{10}$ samples would correspond to the water uptake by particles of different size ranges.

13. Page 19, Line 420: It feels like "*thus smaller particles play a more significant role in $\kappa_{chem}$*" concluded here is a bit too early, as $\kappa_{chem}$ is determined not only by the volume fraction but also by the hygroscopicity of each composition. Small particles with higher $\kappa_{Dp}$ normally correspond to much higher $\kappa$ values for both $\kappa_{chem}$ and $\kappa_{f(RH)}$. Considering the much smaller variation range of $\kappa_{chem}$ or $\kappa_{f(RH)}$ caused by Ångström exponents, influence from $\kappa_{Dp}$ of smaller particles would be more significant.

    Line 427: in Fig.4a, the variation ranges of $\kappa_{chem}$ and $\kappa_{f(RH)}$ are much smaller than those in Fig.4b of PM$_1$. Can we say that the influence of the particle size distribution (as denoted by the Ångström exponent) is not that important for $\kappa_{chem}$ or $\kappa_{f(RH)}$?

14. Page 20, Line 440: "… *with all $\kappa_{chem}$ lower than $\kappa_{f(RH)}$.*" For the discrepancies between $\kappa_{chem}$ and $\kappa_{f(RH)}$, is it also because that $\kappa_{chem}$ is only derived from PM$_{2.5}$ rather than PM$_{10}$ measurements? How to evaluate the effect of size-cut of ACSM especially for measurements with a PM$_{10}$ inlet?

    Line 449: I guess 'NR-PM$_{2.5}$' and 'NR-PM$_1$' is reversely sequenced, similar to the orders of 'PM$_{10}$' and 'PM$_1$'' in the following sentence of the same page and in Line 570 of Page 27.

    Line 456: "*During the first period*", is there any predefinition of the first/second or any other period (e.g., the "non-fog periods" in Page 22, Line 478)?

15. Page 26, Line 555: How is BBOA identified from COA, as the diurnal patterns of the two factors seem to be quite similar?

**Technical corrections:**

1. Page 2 Line 35: "*…correlated with f44 (fraction of m/z 44 in OA)*"

   Line 51: "*…air-pollution-related health effects*"

2. Page 3 Line 54: "*…submicron aerosol particle mass under dry state*"

   Line 70: "*…come from different natural and anthropogenic sources*"

3. Page 4, line 82: "*Studies on $\kappa_{Org}$ have already…*"

   Line 89: "*(HTDMA) or CCN counter was applied…*"

   Line 98: "*…that $\kappa_{Org}$ increases with the increase in particle dry diameter*"

   Line 101: Revise it into "*In this study, the light scattering enhancement factors of both PM$_{10}$ and PM$_1$…*"

4. Page 5, line 115: "*… the diurnal variation of $\kappa_{Org}$ is investigated*"

   Line 117: "*…aerosol chemical compositions measurements is proposed*"

   Line 134: "*…PM impactor inlet, an MFC… and a pump were added*"

5. Page 6, line 138: "*...the first PM$_{10}$ inlet*"

   Line 148: "*chemical composition s of PM$_{2.5}$*"

   Line 150: "*The inlets of group2 and group3 switch every 15 minutes*"

6. Page 7, line 166: "*... after four days of continuous operation (3rd, Dec) and...*"

   Line 174: "*measured RHs / temperatures at the inlet and outlet...*"

7. Page 8, line 196: "*The ion fragments of m/z 38, 49, 63 and 66 were removed*"

8. Page 9, line 222: "*...averaged over each 15−minute observation episode...*"

   Line 223: Change it into "*...and of 30 minutes of SMPS, ACSM ...*"

9. Page 11, line 281: "*...but it(?) is on average 30% lower ...*"

10. Page 12, line 299: "*…loss in semi-volatile  aerosol components. ACSM and the dry nephelometer had a similar tube length and nephelometer measurements bear less*"

*uncertainty than SMPS.*"

Line 305: "*…and volume fractions of the unidentified material*"

Line 309: "*…was not discussed before*"

11. Page 13, Line 321: Delete the comma after 'Eq.2'.

12. Page 14, Line 345: "*… larger $\kappa_{Dp}$ generally corresponded to higher…*"

13. Page 15, Line 352: "*The average PNSD of $PM_{10}$ was applied in the simulation of the…*"

Line 371: "*… $Q_{sca}$ can be expressed as $Q_{sca} = k \cdot D_p$*"

14. Page 18, Line 400: "*(which contribute most to $\sigma_{sp}$ and are the part of the aerosol population that $\kappa_{f(RH)}$ is most sensitive to)*"

Line 407: "*Based on results of Eqs.8 and 20…*"

15. Page 18, Figure 4: It's difficult to tell that what the square/circle stand for? It's preferred to point out briefly in the annotation, instead of just mentioned in the main text. In the annotation, "*Gray areas represent the absolute relative differences between $\kappa_{chem}$ and $\kappa_{f(RH)}$  less than 10%.*".

16. Page 20, Line 453: "*… where quite clean conditions…*"
Line 455: "*…which shows that on average organics contributed most to the mass concentrations of $NR\text{-}PM_1$ and $NR\text{-}PM_{2.5}$*"

17. Page 22, Line 472: "*with relatively small differences…*"

Line 474: "*… they can still cause a small difference…*"

Table 2: "*Average (range) mass concentrations…*". Keep the same format for the names of the five species in the Table, e.g., initials in capitals.

Line 482: "*(see PNSD examples in Fig.S4a)*"

18. Page 23, Line 491: "*was a known parameter… which was  calculated by…*"

Figure 6: The unit of the $f_{OA}$ as shown by the color bar should be %.

Line 494: "*…overestimated the measured one when mass fractions…*"

19. Page 24, Line 514: "*…Period 2 might provide us  a good opportunity…*"

20. Page 25, Line 529: "*hygroscopicity of organic aerosol generally increases as the oxidation level…*"

Line 536: "*The time series of derived $\kappa_{Org}$* *are depicted*"

Line 540: "*…changed from nearly hydrophobic to moderately hygroscopic*"

21. Page 28, Line 581: "*…a constant $\kappa_{Org}$* *was used …*"

Line 585: "*nearly hydrophobic…*"

22. A consistent expression is always recommended in one article, while such inconsistency issues are commonly found in this manuscript. For instance, a subscript format needs to be applied for e.g., (NR-)PM$_1$, (NR-)PM$_{2.5}$, and PM$_{10}$. Different symbols like $\kappa_{fRH}$ and $\kappa_{f(RH)}$ are used randomly. OA is defined as the abbreviation for both 'organic aerosols' and 'organic aerosol'. Please check through the whole content and make corrections in all the necessary places.

23. Some shorten names (such as ACSM, NR-PM) should be defined earlier, i.e., when they appear for the first time.

---

## Referee Comment (RC2) · Anonymous Referee #2 · 15 Nov 2019

Measurements with a humidified nephelometer system and an ACSM were made during the winter on the North China Plain and were used to investigate the hygroscopicity of organic aerosol. The use of f(RH) and bulk chemical composition to calculate kappa for organics is novel. It was found that variability in kappa_org was significantly correlated with the degree of oxidation of the organics. The paper should be published after the concerns below have been addressed. In addition, there is a need for grammatical corrections.

I found the discussion in Section 3.3 to be a little confusing given all of the definitions of kappa. Perhaps a table that lists the different kappas and measurements they are

based on would be helpful.

Line 105: Explain why this diameter range (200 to 800 nm) is represented by the dependence of light scattering on RH.

Figure 1: The text says that the ACSM measured PM2.5 but the figure indicates an upstream cut-off diameter for PM1. Please clarify.

Line 182: Please provide a brief description of the CV and how it allows for the collection of particles as large as 2.5 um.

Lines 185 – 187: Is the CE for the capture vaporizer dependent on chemical composition? Has a unit CE been observed for the composition of the aerosol sampled here?

Figure 4 caption: "...distributions shown in Fig. 4". Should this be Fig. S4?

Lines 449 – 450: Why is the reported maximum PM2.5 concentration less than the PM1 concentration? Same for the PM10 and PM1 light scattering coefficients.

Lines 475 – 478: Does this statement (hygroscopicity of aerosol particles larger than 800 nm is typically lower than for accumulation mode particles) assume uniform composition with size?

---

## Author Comment (AC1) · 6 Dec 2019

**Responses to anonymous referee #1**

**Comment**: Page 3, line 58: "… lead to 40% changes in predicted cloud condensation nuclei (CCN) concentration." The "40% changes" here is confusing, as which is difficult to tell from the sentence whether "a 40% increase or decrease in the NCCN" was resulted from the "50% increase or decrease in $\kappa_{org}$". The similar problem exists in the following sentence, which didn't state clearly the corresponding relationship between the average difference in aerosol radiative forcing and change in $\kappa_{org}$., e.g.,which scenario ($\kappa_{org}$ =0.05 or $\kappa_{org}$ =0.15) corresponds to a higher radiative forcing? A straightforward delivery way is necessary to avoid ambiguity.

**Response**: Thanks for your comment. Changes are made to those sentences to make them more straightforward. The sentence about CCN is modified to "Liu and Wang (2010) demonstrated that 50% increases in $\kappa$ of secondary organic aerosol (0.14±0.07) can result in up to 40% increases in predicted cloud condensation nuclei (CCN) concentration.". The sentence regarding aerosol radiative forcing is revised as "Rastak et al. (2017) reported that global average aerosol radiative forcing could decrease about 1 W/m$^2$ should $\kappa_{OA}$ increase from 0.05 to 0.15".

**Comment**: Page 3, line 65: Unlike the variation of $\kappa_{org}$ itself, I'm afraid I didn't find any connection between the importance of size-dependent $\kappa_{org}$ and the abovementioned content. Some details and corresponding references are needed to support the importance of the size influence on $\kappa_{org}$ and related climatic effects. I would recommend the authors to reorganize the context of size influence on $\kappa_{org}$, which can

be combined with the information provided in the third paragraph (i.e.,contents related to the HTDMA and CCN measurements).

**Response**: Thanks for your comment. We have deleted "size dependence of"

**Comment**: Page 4, line 109: How is the 'mobility diameter of 800 nm' obtained? Related information and references are preferred for the conversion here

**Response**: Thanks for your comment. This part is revised as "particulate matter with aerodynamic diameter less than 1 $\mu m$, corresponding to mobility diameter of approximately 760 nm assuming spherical particles and a particle density of 1.7 g/cm$^3$"

**Comment**: Page 6, line 151: Based on the introduction of each instrument, only the humidified nephelometer can measure both PM10 and PM1. How can chemical compositions of PM10 be measured with ACSM, which is designed with a PM2.5 aerodynamic lens/impactor as mentioned in Line 148? Similarly, how can SMPS measure the size distribution of PM10?

**Response**: Thanks for your comment. As introduced in L152, the inlets of instruments of groups 2 and switch every 15 minutes. This setup makes it possible for ACSM, SMPS and the humidified nephelometer to measure both properties of PM1 and PM10 with a time resolution of 30 minutes. However, the ACSM itself has an impactor with a critical diameter of 2.5 $\mu$m, which is why it cannot measure the total mass of different components of sampled particles when its upstream inlet is PM10.

**Comment**: Page 8, line 206: Why is the density for size conversion regarded as 1.7 g/cm3, the same as that of black carbon used in the calculation of κorg? According to the data reported, the organic fraction is always the predominant contributor to the particle mass of PM2.5. This might suggest a smaller density for the ambient particles. Then how to evaluate the uncertainty in the related calculations?

**Response**: Thanks for your comment. A density of 1.7 g cm3 for the particles larger than 800 nm as a mean density for the coarse mode is a typical value of converting APS aerodynamic diameter to mobility diameter (Wehner et al., 2008). We agree with the reviewer that the aerosol density might change. And the organic fraction is indeed the predominant contributor to the measured particle mass of PM2.5 by ACSM. However, the APS measures the size distribution of coarse particles, and the density of those coarse particles cannot be inferred from or speculated by only using ACSM measurements, because the ACSM cannot measure all components of ambient particles. Some components like dust, which have higher density (Atkinson et al., 2015), cannot be measured by ACSM as discussed in Sec.3.2.

**Comment**: Page 10, line 243: For the "iterative calculation using the Mie theory", how are the chemical composition and corresponding mixing state of particles considered in the κf(RH) calculation? This would affect the closure/comparison between derived κf(RH) and κchem, thus the interpretation of representativeness of κf(RH)

**Response**: The iteration calculation procedure is introduced in Kuang et al. (2017) in detail. The mixing state of particles are assumed to be internally mixed.

**Comment**: Page 11, line 266: I suppose that you were assuming black carbon as hydrophobic, rather than hydrophilic, and $\kappa BC$ is approximately taken as 0. Supporting references would be needed for this point and also for the density assumption of BC in Line 275. A similar typo was found for the description of 'Dust' in Page 12, line 309, which would be hydrophobic instead of hydrophilic.

**Response**: Thanks for your comment. We have changed hydrophilic to hydrophobic for the description of BC and dust. The reference for density assumption of BC was also added.

**Comment**: Page 12, line 307: What does the 'by' mean: "… influences of unidentified material by the ACSM …"? Are you suggesting 'not detected by' ACSM?

**Response**: Yes, it can be understood as "not detected by the ACSM", we think unidentified might be better because those components are indeed sampled in the ACSM but ACSM does not know what they are.

**Comment**: Page 13, line 313: "Bulk aerosol chemical compositions and aerosol hygroscopicity $\kappa f(RH)$ measurements are available, one would naturally jump to the conclusion of treating $\kappa f(RH)$ as $\kappa chem$ to derive $\kappa Org$ (both are from bulk aerosol measurements)."

A connection like a conjunction is needed for the whole sentence. "Both" here sounds ambiguous, although I would assume them to be $\kappa f(RH)$ and $\kappa chem$. It's better

to specific which two hygroscopicity parameters you were referring to.

**Response**: This sentence is modified as "one might naturally jump to the conclusion of treating $\kappa_{f(\mathrm{RH})}$ as $\kappa_{chem}$ to derive $\kappa_{OA}$ because both $\kappa_{f(\mathrm{RH})}$ and $\kappa_{chem}$ are from bulk aerosol measurements"

**Comment**: Page 16, Line 380: Why is the $\kappa f(\mathrm{RH})$ uniform for all the particle sizes? If yes, does it mean that $\kappa Dp = \kappa f(\mathrm{RH})$, while $\kappa Dp$ itself is size dependent?

**Response**: Yes, $\kappa f(\mathrm{RH})$ is defined as the uniform value $\kappa$ that can be used to best fit the observed f(RH). Therefore, $\kappa f(\mathrm{RH})$ being uniform for all the particle sizes is a basic assumption in the $\kappa f(\mathrm{RH})$ retrieval.

**Comment**: Page 17, Line 387: What does the 'which' mean?

Line 388: Is there any consideration of choosing "two extreme cases of size resolved $\kappa Dp$

**Response**: "which" is changed to "this result of $X_c$" to make it clearer. The relative difference between $\kappa f(\mathrm{RH})$ and $\kappa$chem are mostly influenced by shape of size-resolved $\kappa Dp$ distribution. Thus, the two extreme cases of size resolved $\kappa Dp$ can give the upper range of relative differences of $\kappa f(\mathrm{RH})$ and $\kappa$chem for PM1.

**Comment**: Page 18, Figure 4(c): In comparison of the PM1 scenario displayed in Fig.4b, $\kappa f(\mathrm{RH})$ is generally higher than $\kappa chem$, and larger discrepancies exist for the PM10

case. Can you provide some hints for these results?

Line 411: "How much does $\kappa chem$ differ from $\kappa f(RH)$ for PM1 and PM10 samples?"

In my understanding, the $\kappa chem$ of PM10 samples is calculated from the corresponding chemical compositions that are actually measured for PM2.5 instead of PM10 (due to the configuration of ACSM with a PM2.5 impactor), when sampling with a PM10 inlet. In this sense, the two hygroscopicity parameters for PM10 samples would correspond to the water uptake by particles of different size ranges.

**Response**: The chemical component measurements during the field campaign used in this study is not used to discuss differences of $\kappa chem$ and $\kappa f(RH)$ of PM10. The $\kappa chem$ and $\kappa f(RH)$ of PM10 shown in Fig.4c are calculated based on size-resolved $\kappa$ distribution as shown in Fig.S5. To make this part clearer, the paragraph describing results of Fig.4c is revised as the following: "For PM10, values of $\kappa_{chem}$ and $\kappa_{f(RH)}$ using $\kappa_{D_p}$ size distributions derived from ambient measurements (Fig.S5, similar to Fig.4b) were simulated and displayed in Fig.4c. The simulated absolute values of the relative difference between $\kappa_{chem}$ and $\kappa_{f(RH)}$ ranged from 0.2% to 41% with an average and standard deviation of 16±8 %, with all $\kappa_{chem}$ lower than $\kappa_{f(RH)}$. This is because, for PM10, super-micron particles typically with low hygroscopicity (Fig.S5) contribute much more to $V_{tot}$ than to $\sigma_{sp}$ (as shown in Fig.S7). These results indicate that, for PM10, $\kappa_{f(RH)}$ cannot accurately represent $\kappa_{chem}$."

"

**Comment**: Page 19, Line 420: It feels like "thus smaller particles play a more

significant role in $\kappa chem$" concluded here is a bit too early, as $\kappa chem$ is determined not only by the volume fraction but also by the hygroscopicity of each composition. Small particles with higher $\kappa Dp$ normally correspond to much higher $\kappa$ values for both $\kappa chem$ and $\kappa f(RH)$. Considering the much smaller variation range of $\kappa chem$ or $\kappa f(RH)$ caused by Ångström exponents, influence from $\kappa Dp$ of smaller particles would be more significant.

Line 427: in Fig.4a, the variation ranges of $\kappa chem$ and $\kappa f(RH)$ are much smaller than those in Fig.4b of PM1. Can we say that the influence of the particle size distribution (as denoted by the Ångström exponent) is not that important for $\kappa chem$ or $\kappa f(RH)$?

**Response**: We think the reviewer has a misunderstanding here. This sentence is to explain why calculated $\kappa_{chem}$ is smaller than $\kappa_{fRH}$ for the results shown in Fig.4a. And the variation ranges of $\kappa chem$ and $\kappa f(RH)$ in Fig.4a are much smaller than those in Fig.4b of PM1, because $\kappa chem$ and $\kappa f(RH)$ in Fig.4a are calculated based on fixed size-resolved $\kappa Dp$ distribution shown in Fig.3. The $\kappa chem$ and $\kappa f(RH)$ shown in Fig.4b are calculated based on about 23 size-resolved $\kappa Dp$ distribution which are derived from measured size-resolved chemical compositions in the NCP region. Based on the results of Fig.4, we can say that the influence of particle size distribution (as denoted by the Ångström exponent) is not that important for $\kappa chem$ or $\kappa f(RH)$.

**Comment**: Page 20, Line 440: "… with all $\kappa chem$ lower than $\kappa f(RH)$." For the discrepancies between $\kappa chem$ and $\kappa f(RH)$, is it also because that $\kappa chem$ is only derived

from PM2.5 rather than PM10 measurements? How to evaluate the effect of size-cut of ACSM especially for measurements with a PM10 inlet?

Line 449: I guess 'NR-PM2.5' and 'NR-PM1' is reversely sequenced, similar to the orders of 'PM10' and 'PM1'' in the following sentence of the same page and in Line 570 of Page 27

Line 456: "During the first period", is there any predefinition of the first/second or any other period (e.g., the "non-fog periods" in Page 22, Line 478)?

**Response**: Thanks for your comment. This part is still theoretically discussing the discrepancies between $\kappa chem$ and $\kappa f(RH)$, both kchem and $\kappa f(RH)$ are derived from size-resolved $\kappa$ distribution of PM10. Discussions here are not relevant to ACSM measurements.

'NR-PM2.5' and 'NR-PM1' are indeed reversely sequenced, and we have revised these sentences. We have changed "During the first period" to "During the period 1 shown in Fig.5" to make it clearer. The "non-fog periods" are changed to "non-fog periods (periods with RH <100%)."

**Comment**: Page 26, Line 555: How is BBOA identified from COA, as the diurnal patterns of the two factors seem to be quite similar?

**Response**:

We thank the reviewer for this comment. Although the diurnal profiles of BBOA and COA have some similarities, the mass spectra and temporal variations of the two factors were different (Figure R1). In particular, the BBOA spectrum was characterized by

pronounced *m/z* 60, a tracer *m/z* for biomass burning due to fragmentation of levoglucosan (Cubison et al., 2011), and the spectrum of COA showed much higher *m/z* 55/57 ratio that has been widely used as a diagnostic of cooking emissions (Mohr et al., 2012). Because of the spectral differences between BBOA and COA, these two factors can be well separated by positive matrix factorization (PMF).

[Figure]

Figure R1. Mass spectra and time series of five OA factors that were identified in this study.

Cubison, M. J., Ortega, A. M., Hayes, P. L., Farmer, D. K., Day, D., Lechner, M. J., Brune, W. H., Apel, E., Diskin, G. S., Fisher, J. A., Fuelberg, H. E., Hecobian, A., Knapp, D. J., Mikoviny, T., Riemer, D., Sachse, G. W., Sessions, W., Weber, R. J., Weinheimer, A. J., Wisthaler, A., and Jimenez, J. L.: Effects of aging on organic aerosol from open biomass burning smoke in aircraft and laboratory studies, Atmos. Chem. Phys., 11, 12049-12064, 10.5194/acp-11-12049-2011, 2011.

Mohr, C., DeCarlo, P. F., Heringa, M. F., Chirico, R., Slowik, J. G., Richter, R., Reche, C., Alastuey, A., Querol, X., Seco, R., Peñuelas, J., Jiménez, J. L., Crippa, M., Zimmermann, R., Baltensperger, U., and Prévôt, A. S. H.: Identification and quantification of organic aerosol from cooking and other sources in Barcelona using aerosol mass spectrometer data, Atmos. Chem. Phys., 12, 1649-1665, 10.5194/acp-12-1649-2012, 2012.

**Technical Corrections**

**Comment**: A consistent expression is always recommended in one article, while such inconsistency issues are commonly found in this manuscript. For instance, a subscript format needs to be applied for e.g., (NR-)PM1, (NR-)PM2.5, and PM10. Different symbols like κfRH and κf(RH) are used randomly. OA is defined as the abbreviation for both 'organic aerosols' and 'organic aerosol'. Please check through the whole content and make corrections in all the necessary places.

Some shorten names (such as ACSM, NR-PM) should be defined earlier, i.e., when they appear for the first time.

**Response**: Corrections have been made according to the suggestions. Subscripts were applied for (NR-)PM$_1$, (NR-)PM$_{2.5}$, and PM$_{10}$. All κfRH were changed to κf(RH), and shortened names like ACSM, NR-PM were defined when they appear for the first time.

**Reference:**

Atkinson, D.B., Radney, J.G., Lum, J., Kolesar, K.R., Cziczo, D.J., Pekour, M.S., Zhang, Q., Setyan, A., Zelenyuk, A., Cappa, C.D., 2015. Aerosol optical hygroscopicity measurements during the 2010 CARES campaign. Atmospheric Chemistry and Physics 15, 4045-4061.

Kuang, Y., Zhao, C., Tao, J., Bian, Y., Ma, N., Zhao, G., 2017. A novel method for deriving the aerosol hygroscopicity parameter based only on measurements from a humidified nephelometer system. Atmos. Chem. Phys. 17, 6651-6662.

Wehner, B., Birmili, W., Ditas, F., Wu, Z., Hu, M., Liu, X., Mao, J., Sugimoto, N., Wiedensohler, A., 2008. Relationships between submicrometer particulate air pollution and air mass history in Beijing, China, 2004–2006. Atmos. Chem. Phys. 8, 6155-6168.

---

## Author Comment (AC2) · 6 Dec 2019

**Responses to anonymous referee #2**

**Specific comments**

**Comment**: I found the discussion in Section 3.3 to be a little confusing given all of the definitions of kappa. Perhaps a table that lists the different kappas and measurements they are based on would be helpful.

**Response**: Thanks for your suggestion. A Table as the following was added to improve the readability of the manuscript.

Table 2. Different $\kappa$ and their physical meanings

| | |
|---|---|
| $\kappa_{f(\mathrm{RH})}$ | A uniform $\kappa$ for all particle sizes which describes $f(\mathrm{RH})$ accurately |
| $\kappa_{chem}$ | A bulk $\kappa$ assuming different chemical compositions of aerosol populations are internally mixed and calculated with the ZSR mixing rule |
| $\kappa_i$ | hygroscopicity parameter $\kappa$ of chemical species i |
| $\kappa_{D_p}$ | The $\kappa$ assuming different chemical compositions of particles with diameter of $D_p$ are internally mixed and calculated with the ZSR mixing rule |

**Comment**: Line 105: Explain why this diameter range (200 to 800 nm) is represented by the dependence of light scattering on RH.

**Response**: This is too complex to be explained in a few sentences within the Introduction part, which is why detailed explanations are given in Sect3.3.

**Comment**: Figure 1: The text says that the ACSM measured PM2.5 but the figure indicates an upstream cut-off diameter for PM1. Please clarify.

**Response**: As introduced in Sect.2.1 and shown in Fig.1, the upstream impactor of

ACSM switches between PM1 and PM10 every 15 minutes.

**Comment**: Line 182: Please provide a brief description of the CV and how it allows for the collection of particles as large as 2.5 um.

**Response**: Thank the reviewer's comments. In the revised manuscript, we expanded the description of CV. It now reads: "The CV was designed with an enclosed cavity to increase particle collection efficiency (CE) at the detector (Xu et al., 2017). Both laboratory and field measurements indicate that the CE of CV was fairly robust and was roughly equivalent to 1. Therefore, a CE of 1 was applied to all measured species in this study (Hu et al., 2017; Hu et al., 2018b)."

Aerodyne Research Inc. redesigned the aerodynamic lens by changing the geometry of the exit nozzle (Xu et al., 2017). Compared with the traditional $PM_1$ standard lens, the transmission efficiency of the new lens is about 50% at 3.5 μm vacuum aerodynamic diameter, which is approximately equal to a 2.8 μm aerodynamic diameter assuming an average ambient particle density of 1.7 $g/cm^3$. Therefore, the new aerodynamic lens allows for the collection of particles as large as 2.5 μm.

**Comment**: Lines 185 – 187: Is the CE for the capture vaporizer dependent on chemical composition? Has a unit CE been observed for the composition of the aerosol sampled here?

**Response**: The collection efficiency for the capture vaporizer (CV) is independent of

chemical composition. Hu et al. (2017) evaluated comprehensively the CV in three field studies, and found that the CE of CV was fairly robust and was roughly equivalent to 1. Therefore, a CE of 1 was applied to all measured species in this study.

**Comment:** L126: Figure 4 caption: "…distributions shown in Fig. 4". Should this be Fig. S4?

**Response:** Thank you for noticing. We have changed Fig. 4 as Fig.S5.

**Comment**: Lines 449 – 450: Why is the reported maximum PM2.5 concentration less than the PM1 concentration? Same for the PM10 and PM1 light scattering coefficients.

**Response**: Thank you for noticing, these are typing errors, it should be the other way around.

**Reference:**

Hu, W., Campuzano-Jost, P., Day, D. A., Croteau, P., Canagaratna, M. R., Jayne, J. T., Worsnop, D. R., and Jimenez, J. L.: Evaluation of the new capture vaporizer for aerosol mass spectrometers (AMS) through field studies of inorganic species, Aerosol Sci. Tech., 51, 735–754, 10.1080/02786826.2017.1296104, 2017.

---

## Author Response (AR2)

**Dear Editor:**

Thank you very much for your attention and consideration. Authors aand affiliations are made consistent between the manuscript and the supplement file. The data availability part are modified as:

**Data availability**. The data used in this study are listed in the references and a repository at https://pan.baidu.com/s/16dOPuTQ568z5JRGF0jGLHQ (both python and matlab format), and also available from the corresponding author upon request (kuangye@jnu.edu.cn) and (sunyele@mail.iap.ac.cn).

"

Sincerely Yours

Ye Kuang and Yele Sun